# THE GEOMETRY OF SIGN GRADIENT DESCENT

## ABSTRACT

Sign gradient descent has become popular in machine learning due to its favorable communication cost in distributed optimization and its good performance in neural network training. However, we currently do not have a good understanding of which geometrical properties of the objective function determine the relative speed of sign gradient descent compared to standard gradient descent. In this work, we frame sign gradient descent as steepest descent with respect to the maximum norm ($L^\infty$-norm). We review the steepest descent framework and the related concept of smoothness with respect to arbitrary norms. By studying the smoothness constant resulting from the $L^\infty$-geometry, we isolate properties of the objective which favor sign gradient descent relative to gradient descent. In short, we find two requirements on its Hessian: (i) some degree of "diagonal dominance" and (ii) the maximal eigenvalue being much larger than the average eigenvalue. We also clarify the meaning of a certain separable smoothness assumption used in previous analyses of sign gradient descent. Experiments verify the developed theory.

## 1 INTRODUCTION

We consider an unconstrained, continuous optimization problem, $\min_{\boldsymbol{x} \in \mathbb{R}^d} f(\boldsymbol{x})$, with a differentiable and lower-bounded objective $f \colon \mathbb{R}^d \to \mathbb{R}$. The prototypical first-order optimization algorithm to solve this problem is gradient descent (GD), which iteratively updates $\boldsymbol{x}_{t+1} = \boldsymbol{x}_t - \alpha_t \nabla f_t$, where $f_t := f(\boldsymbol{x}_t)$. Several recent works have considered the sign gradient descent (signGD) method,

$$\boldsymbol{x}_{t+1} = \boldsymbol{x}_t - \alpha_t \operatorname{sign}(\nabla f_t)\,, \tag{1}$$

where the application of the sign is to be understood elementwise. This method is attractive in distributed optimization where it conveniently reduces the communication cost to a single bit per gradient coordinate (e.g., Seide et al., 2014; Karimireddy et al., 2019), but its interest extends beyond that. Balles & Hennig (2018) and Bernstein et al. (2018) point out that sign gradient descent is closely related to the popular Adam method (Kingma & Ba, 2015) and demonstrate that it often achieves similar practical performance on deep learning tasks. Studying sign gradient descent can thus be seen as a step towards a better understanding of the ubiquitous Adam method.

### 1.1 OVERVIEW AND CONTRIBUTIONS

In this work, we make the following contributions:

1. We review the concept of steepest descent with respect to arbitrary norms, which includes both gradient descent and sign gradient descent. While these ideas feature in previous works, we provide a concise and general write-up which we hope to be useful in its own right. We emphasize the role of the smoothness constant of $f$ w.r.t. a norm as a major driver of the performance of the associated steepest descent method.

2. Since sign gradient descent is steepest descent w.r.t. the maximum norm, we study the corresponding smoothness constant $L_\infty$, isolating properties of the objective function which favor sign gradient descent. We pinpoint two requirements on its Hessian: ((i) some degree of "diagonal dominance" and (ii) the maximal eigenvalue being much larger than the average eigenvalue.

3. We verify our theory experimentally on both quadratic functions and deep networks.

| Method | Norm | Dual | Update direction | |
|---|---|---|---|---|
| Gradient descent | $\|\cdot\|_2$ | $\|\cdot\|_2$ | $\nabla f$ | |
| Sign gradient descent | $\|\cdot\|_\infty$ | $\|\cdot\|_1$ | $\|\nabla f\|_1 \operatorname{sign}(\nabla f)$ | |
| Coordinate descent | $\|\cdot\|_1$ | $\|\cdot\|_\infty$ | $\nabla f_{i_{\max}} e^{(i_{\max})}$ | (see Appendix. A.1) |
| Block-normalized GD | $\|\cdot\|_{\mathcal{B},\infty}$ | $\|\cdot\|_{\mathcal{B},1}$ | $\|\nabla f\|_{\mathcal{B},1} \operatorname{norm}_{\mathcal{B}}(\nabla f)$ | (see Appendix. A.1) |

Table 1: A few steepest descent methods. The table lists the used norm $\|\cdot\|$, its dual $\|\cdot\|_*$, and the resulting update direction. The significance of the dual norm will be explained in §2.1.

## 1.2 RESTRICTIONS

While modern machine learning often relies on *stochastic* optimization, this work focuses on the *geometric* aspects of sign gradient descent and we restrict ourselves to the deterministic setting to facilitate this analysis. Furthermore, our interest in signGD is from an optimization perspective and we will not consider the question of generalization. It should be noted that sign gradient descent has been observed to lead to slightly deteriorating generalization performance compared to gradient descent in neural network training (Wilson et al., 2017; Balles & Hennig, 2018).

## 2 STEEPEST DESCENT AND SMOOTHNESS CONSTANTS

In this section, we review the steepest descent[1] framework, which encompasses both gradient descent and sign gradient descent. Embedding both methods in a common framework will facilitate our comparative analysis. Steepest descent with respect to a norm $\|\cdot\|$ is defined via the update rule

$$\boldsymbol{x}_{t+1} \in \underset{\boldsymbol{x}\in\mathbb{R}^d}{\arg\min} \left( f_t + \langle \nabla f_t, \boldsymbol{x} - \boldsymbol{x}_t \rangle + \frac{1}{2\alpha_t}\|\boldsymbol{x} - \boldsymbol{x}_t\|^2 \right), \tag{2}$$

minimizing a local linear approximation of $f$ plus a quadratic norm penalty. This minimizer need not be unique, in which case steepest descent is to be understood as choosing *any* solution.

It is straight-forward to see that gradient descent is steepest descent with respect to the Euclidean norm. Similarly, a version of sign gradient descent arises as steepest descent with respect to the maximum norm, $\|\boldsymbol{z}\|_\infty := \max_i |z_i|$. Namely,

$$\boldsymbol{x}_t - \alpha_t\|\nabla f_t\|_1 \operatorname{sign}(\nabla f_t) \in \underset{\boldsymbol{x}\in\mathbb{R}^d}{\arg\min} \left( f_t + \langle \nabla f_t, \boldsymbol{x} - \boldsymbol{x}_t \rangle + \frac{1}{2\alpha_t}\|\boldsymbol{x} - \boldsymbol{x}_t\|_\infty^2 \right). \tag{3}$$

This update is equivalent to Eq. (1) up to the scaling with the gradient norm $\|\nabla f_t\|_1$ which may be subsumed in the step size.[2] Table 1 lists a few more steepest descent methods, which are discussed in more detail in Appendix A.1.

In the following, we will explore steepest descent in more detail by a) relating it to the concept of smoothness w.r.t. arbitrary norms; b) tying its convergence speed to the corresponding smoothness constant; c) linking the smoothness constants to properties of the Hessian.

## 2.1 SMOOTHNESS WITH RESPECT TO ARBITRARY NORMS

Steepest descent is closely connected to the concept of smoothness w.r.t. the relevant norm. We recall that a function $f$ is $L$-smooth w.r.t. a norm $\|\cdot\|$ if

$$\|\nabla f(\boldsymbol{y}) - \nabla f(\boldsymbol{x})\|_* \leq L\|\boldsymbol{y} - \boldsymbol{x}\|, \quad \forall \boldsymbol{x}, \boldsymbol{y} \in \mathbb{R}^d, \tag{4}$$

where $\|\cdot\|_*$ denotes the dual norm of $\|\cdot\|$, defined as $\|\boldsymbol{z}\|_* := \max_{\|\boldsymbol{x}\|\leq 1}\langle \boldsymbol{z}, \boldsymbol{x}\rangle$. Table 1 lists the dual norm pairs for the methods under consideration in this paper. Smoothness directly motivates steepest descent via the following quadratic bound on the function:

**Lemma 1.** *If $f$ is $L$-smooth w.r.t. $\|\cdot\|$, then $f(\boldsymbol{y}) \leq f(\boldsymbol{x}) + \langle\nabla f(\boldsymbol{x}), \boldsymbol{y}-\boldsymbol{x}\rangle + \frac{L}{2}\|\boldsymbol{y}-\boldsymbol{x}\|^2 \ \forall \boldsymbol{x}, \boldsymbol{y} \in \mathbb{R}^d$.*

---

[1] Also known as *generalized* or *non-Euclidean* gradient descent (Carlson et al., 2015; Kelner et al., 2014).

[2] In this paper we study this version of sign gradient descent to facilitate the analysis in the steepest descent framework. In practice it is more common to use Eq. (1) with a constant or manually decreasing step size. In Appendix C, we discuss a possible explanation for this discrepancy.

Setting $\boldsymbol{x} = \boldsymbol{x}_t$ we get $f(\boldsymbol{y}) \leq f_t + \langle \nabla f_t, \boldsymbol{y} - \boldsymbol{x}_t \rangle + \frac{L}{2} \|\boldsymbol{y} - \boldsymbol{x}_t\|^2$ which resembles Eq. (2) and minimizing with respect to $\boldsymbol{y}$ results in a steepest descent update with step size $\alpha_t = 1/L$. (The proofs of this lemma and all subsequent results are provided in Appendix E.)

Due to the equivalence of norms on $\mathbb{R}^d$, a function that is smooth with respect to one norm is also smooth with respect to any other norm. However, the tightest possible smoothness constant,

$$L := \sup_{\boldsymbol{x} \neq \boldsymbol{y}} \frac{\|\nabla f(\boldsymbol{y}) - \nabla f(\boldsymbol{x})\|_*}{\|\boldsymbol{y} - \boldsymbol{x}\|}, \tag{5}$$

will differ depending on the choice of norm. In the following, when we say $f$ is $L$-smooth w.r.t. $\|\cdot\|$, we will always assume $L$ to be given by Eq. (5). As we will see next, this constant will govern the convergence speed of the corresponding steepest descent method.

## 2.2 Convergence of Steepest Descent

We now recall existing convergence results for steepest descent methods based on the smoothness constants w.r.t. arbitrary norms. The assumption of smoothness is crucial since it guarantees an improvement in function value:

**Lemma 2.** *If $f$ is $L$-smooth w.r.t. $\|\cdot\|$, then a single steepest descent step with step size $1/L$ satisfies*

$$f(\boldsymbol{x}^+) \leq f(\boldsymbol{x}) - \frac{1}{2L} \|\nabla f(\boldsymbol{x})\|_*^2. \tag{6}$$

Without additional assumptions, this yields $O(1/T)$ convergence to a first-order stationary point:

**Proposition 1.** *If $f$ is $L$-smooth w.r.t. $\|\cdot\|$, then steepest descent (Eq. 2) with $\alpha_t \equiv 1/L$ satisfies*

$$\frac{1}{T} \sum_{t=0}^{T-1} \|\nabla f_t\|_*^2 \leq \frac{2L(f_0 - f^\star)}{T}. \tag{7}$$

By nature of Proposition 1, all steepest descent methods will enjoy the same rate of convergence. But Lemma 2 and Proposition 1 differ across different steepest descent methods in (i) the smoothness constant, and (ii) the norm in which the gradient magnitude is measured. These two aspects will play a role when we compare sign gradient descent and gradient descent in Section 4.

Appendix A contains stronger convergence results under additional assumptions, in particular linear convergence under the Polyak-Łojasiewicz condition. We omit these results here as they are not of primary interest to our discussion and, in fact, hide the dependence on the gradient norm which we see clearly in Lemma 2 and Proposition 1.

## 2.3 Smoothness as a Bound on the Hessian

The definition of smoothness in Eq. (4) is unwieldy. For the Euclidean norm, a more intuitive characterization is widely known: a twice-differentiable function $f$ is $L_2$-smooth w.r.t. the Euclidean norm if and only if $\|\nabla^2 f(\boldsymbol{x})\|_2 \leq L_2$ for all $\boldsymbol{x} \in \mathbb{R}^d$. Here, $\|\cdot\|_2$ for matrices denotes the spectral norm, given by the largest-magnitude eigenvalue for symmetric matrices. In the following proposition we show that smoothness with respect to any other norm likewise arises from a bound on the Hessian, but in different matrix norms. In Section 3, we will use this to better understand the smoothness constant w.r.t. the maximum norm, which critically affects the performance of signGD.

**Proposition 2.** *For any norm $\|\cdot\|$ on $\mathbb{R}^d$, we define the matrix norm*

$$\|\boldsymbol{H}\| := \sup_{\|\boldsymbol{x}\| \leq 1} \|\boldsymbol{H}\boldsymbol{x}\|_* . \tag{8}$$

*Then, a twice-differentiable $f$ is $L$-smooth w.r.t. $\|\cdot\|$ if and only if $\|\nabla^2 f(\boldsymbol{x})\| \leq L$ for all $\boldsymbol{x}$.*

The proof can be found in Appendix E; we are not aware of prior works showing this for the general case. For the Euclidean norm, Proposition 2 gives back the familiar spectral norm bound.

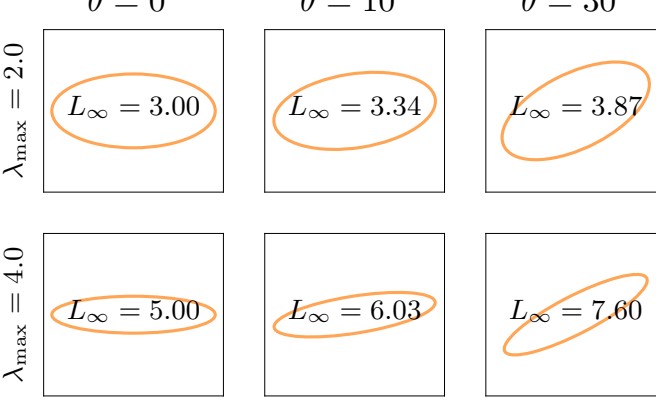

Figure 1: $L_\infty$ *increases as the Hessian becomes less axis-aligned.* Contour line of $f(\boldsymbol{x}) = \frac{1}{2}\boldsymbol{x}^T\boldsymbol{H}\boldsymbol{x}$, which forms an ellipse with principal axes given by the eigenvectors and axis lengths given by the inverse eigenvalues. We construct $\boldsymbol{H} \in \mathbb{R}^{d \times d}$ with a fixed $\lambda_{\min} = 1$ and variable $L_2 = \lambda_{\max} > 1$ as well as the angle $\theta$ between eigenvectors and coordinate axes.

## 3 UNDERSTANDING THE $L_\infty$-SMOOTHNESS CONSTANT

We have seen in Section 2.2 that the smoothness constant w.r.t. a norm is a crucial driver of the convergence speed of the associated steepest descent method. While we have a good intuition for Euclidean smoothness—an upper bound on the eigenvalues of the Hessian—this is somewhat lacking for smoothness w.r.t. the maximum norm. Our goal in this section is to understand which properties of the objective function affect the constant $L_\infty$. The Euclidean smoothness constant $L_2$ will thereby serve as a reference point. We assume $f$ to be $L_2$-smooth w.r.t. $\|\cdot\|_2$ and $L_\infty$-smooth w.r.t. $\|\cdot\|_\infty$ in the "tight" sense of Eq. (5). Basic inequalities between the norms yield (see Appendix E)

$$L_2 \le L_\infty \le dL_2 . \tag{9}$$

The smoothness constant governing the convergence speed of sign gradient descent will always be larger than the one pertinent to gradient descent. This does *not* mean that signGD is always worse than GD. In addition to being a bound, the smoothness constant is only one factor influencing the convergence speed; Lemma 2 shows that the dual gradient norm plays a role as well. As we will discuss in Section 4, this gradient norm is larger for signGD which can make up for the disadvantage of a larger smoothness constant. In this section we will characterize conditions under which $L_\infty$ is "not too bad"—in particular much smaller than its worst case of $dL_2$—such that signGD can be competitive with gradient descent.

Our discussion will use the Hessian-based formulation of the smoothness constants (Proposition 2). We assume $f$ to be twice-differentiable. As mentioned before, the Euclidean smoothness constant is simply given by the largest-magnitude eigenvalue. The max-norm smoothness constant is determined by

$$L_\infty = \sup_{\boldsymbol{x} \in \mathbb{R}^d} \|\nabla^2 f(\boldsymbol{x})\|_{\infty,1}, \quad \|\boldsymbol{H}\|_{\infty,1} := \sup_{\|\boldsymbol{x}\|_\infty \le 1} \|\boldsymbol{H}\boldsymbol{x}\|_1. \tag{10}$$

Unfortunately, computing $\|\boldsymbol{H}\|_{\infty,1}$ is NP-hard (Rohn, 2000). However, we can obtain some insight through the following upper bound.

**Proposition 3.** *Let $\boldsymbol{H} \in \mathbb{R}^{d \times d}$ be nonzero, positive semi-definite with eigenvalues $\lambda_1, \ldots, \lambda_d$. Then*

$$\|\boldsymbol{H}\|_{\infty,1} \le \rho_{diag}(\boldsymbol{H})^{-1} \sum_i \lambda_i \quad \text{with} \quad \rho_{diag}(\boldsymbol{H}) := \frac{\sum_i |H_{ii}|}{\sum_{i,j} |H_{ij}|} . \tag{11}$$

The quantity $\rho_{\mathrm{diag}}(\boldsymbol{H}) \in [d^{-1}, 1]$ measures the percentage of diagonal mass of the Hessian and thus relates to the "axis-alignment" of the objective. Indeed, diagonal matrices satisfy $\rho_{\mathrm{diag}} = 1$ and their eigenvectors coincide with the coordinate axes. Proposition 3 thus indicates that $L_\infty$ will depend both on the "axis-alignment" of the Hessian as well as its eigenvalues. This is in stark contrast to gradient descent, for which the relevant matrix norm, $\|\boldsymbol{H}\|_2$, is invariant to rotations. Figure 1 provides a two-dimensional illustration.

Using Proposition 3, we can now answer our question: Under what conditions is $L_\infty$ "not too bad" compared to $L_2$? Firstly, $L_\infty$ benefits from high $\rho_{\mathrm{diag}}$, i.e., an **approximately diagonal Hessian or**

**axis-aligned objective**. Secondly, the sum of eigenvalues should be much smaller than its worst case of $d\lambda_{\max}$ or, equivalently, **the average eigenvalue should be much smaller than the maximal one**: $\bar{\lambda} := \frac{1}{d}\sum_i \lambda_i \ll \lambda_{\max}$. The second condition will be met if the spectrum is dominated by a small number of "outlier eigenvalues" that far exceed the bulk of eigenvalues. Notably, both the relative axis-alignment and the skewed spectrum have been observed in neural network training (Adolphs et al., 2019; Chaudhari et al., 2017; Ghorbani et al., 2019).

The discussion in the preceding paragraphs was based on an upper bound, since $\|\boldsymbol{H}\|_{\infty,1}$ is generally intractable. Section 5 will feature experiments on quadratics of moderate dimension, where $\|\boldsymbol{H}\|_{\infty,1}$ can be computed to confirm these findings for the exact $L_\infty$.

## 4  GRADIENT DESCENT VS SIGN GRADIENT DESCENT

We found in Section 3 that $L_\infty$ will always exceed $L_2$, and identified conditions under which $L_\infty \ll dL_2$. By Lemma 2, the guaranteed improvement of the two methods at $\boldsymbol{x} \in \mathbb{R}^d$ is

$$\mathcal{I}_{\text{GD}}(\boldsymbol{x}) := \frac{\|\nabla f(\boldsymbol{x})\|_2^2}{L_2} \quad \text{vs.} \quad \mathcal{I}_{\text{signGD}}(\boldsymbol{x}) := \frac{\|\nabla f(\boldsymbol{x})\|_1^2}{L_\infty}.$$

We will now compare the two methods based on these improvement guarantees, thus taking into account the dual gradient norm in addition to the smoothness constant. Basic inequalities between the two relevant norms, $\sqrt{d}\|\boldsymbol{z}\|_2 \geq \|\boldsymbol{z}\|_1 \geq \|\boldsymbol{z}\|_2$ for all $\boldsymbol{z} \in \mathbb{R}^d$, show that this potentially favors sign gradient descent. Following Bernstein et al. (2018), it will be convenient to define the ratio

$$\phi(\boldsymbol{z}) := \frac{\|\boldsymbol{z}\|_1^2}{d\|\boldsymbol{z}\|_2^2} \in [d^{-1}, 1],$$

which can be seen as a measure of the "density" of the vector $\boldsymbol{z}$, i.e., how evenly its mass is distributed across coordinates.[3] With that, we can write

$$\mathcal{R}(\boldsymbol{x}) := \frac{\mathcal{I}_{\text{signGD}}(\boldsymbol{x})}{\mathcal{I}_{\text{GD}}(\boldsymbol{x})} = \frac{\|\nabla f(\boldsymbol{x})\|_1^2}{\|\nabla f(\boldsymbol{x})\|_2^2} \frac{L_2}{L_\infty} = \phi(\nabla f(\boldsymbol{x})) \frac{dL_2}{L_\infty}. \tag{12}$$

If $L_\infty \ll dL_2$, for which we have identified conditions in Section 3, and the gradient is dense, i.e. $\phi(\nabla f(\boldsymbol{x})) \gg d^{-1}$, then sign gradient descent will achieve a larger improvement than gradient descent at $\boldsymbol{x} \in \mathbb{R}^d$.

We can make this comparison even more palpable. We restrict to quadratic functions, such that $\nabla^2 f(\boldsymbol{x}) \equiv \boldsymbol{H}$ and $L_\infty = \|\boldsymbol{H}\|_{\infty,1}$ as well as $L_2 = \|\boldsymbol{H}\|_2 = \lambda_{\max}$. Assume $\boldsymbol{H}$ to be positive definite with eigenvalues $\lambda_1, \ldots, \lambda_d$. Then Proposition 3 in Eq. (12) yields

$$\mathcal{R}(\boldsymbol{x}) = \frac{\mathcal{I}_{\text{signGD}}(\boldsymbol{x})}{\mathcal{I}_{\text{GD}}(\boldsymbol{x})} \geq \underbrace{\rho_{\text{diag}}(\boldsymbol{H})}_{\in [d^{-1},1]} \underbrace{\frac{\lambda_{\max}}{\bar{\lambda}}}_{\in [1,d]} \underbrace{\phi(\nabla f(\boldsymbol{x}))}_{\in [d^{-1},1]} \tag{13}$$

Hence, sign gradient descent will outperform gradient descent if (i) the Hessian is to some extent diagonal, (ii) the maximal eigenvalue is much larger than the average eigenvalue, and (iii) the gradient vector is relatively dense.

We emphasize that this is a *local* comparison at $\boldsymbol{x} \in \mathbb{R}^d$ due to the dependence on the gradient $\nabla f(\boldsymbol{x})$. It is tempting to try and lower-bound the gradient density $\phi(\nabla f(\boldsymbol{x})) \geq \phi_g \gg d^{-1}$ for all $\boldsymbol{x}$ in order to make global statements. Bernstein et al. (2018) assume such a lower bound when contrasting signGD and GD in their analysis. However, $\phi(\nabla f(\boldsymbol{x}))$ can easily be shown to attain $d^{-1}$ even on quadratics. Any non-trivial lower bound would thus have to be restricted to the trajectory of the optimizer and take into account the initialization, which seems out of reach at the moment. The effect of the gradient norm thus remains an empirical question which we now address.

## 5  EXPERIMENTS

### 5.1  QUADRATIC PROBLEMS

We first consider synthetic quadratic problems of moderate dimension $d = 12$, where we can compute and control all relevant quantities. To validate our theoretical findings, we generate Hessians

---

[3]Note that $\phi(\boldsymbol{z}) = 1$ for $\boldsymbol{z} \propto (1, \ldots, 1)^T$ and $\phi(\boldsymbol{z}) = d^{-1}$ for $\boldsymbol{z} \propto (1, \ldots, 0)^T$.

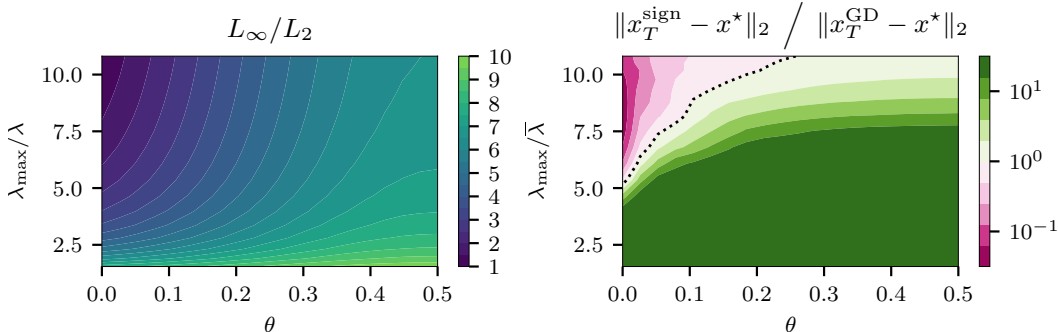

Figure 2: We consider quadratic objectives varying across two axes: $\lambda_{\max}/\overline{\lambda}$ as well as a rotation value $\theta$. The left plot depicts the ratio of the two relevant smoothness constants. $L_\infty$ is sensitive to $\theta$ and grows relative to $L_2 = \lambda_{\max}$ as the problem becomes less axis-aligned. The right plot depicts the relative performance of gradient descent and sign gradient descent on these problems. GD drastically (the colormap is clipped) outperforms signGD for mildly conditioned (small $\lambda_{\max}/\overline{\lambda}$) and non-axis-aligned (large $\theta$) problems. However, sign gradient descent is preferable for problems with high $\lambda_{\max}/\overline{\lambda}$, given that they have some degree of axis-alignment (small $\theta$). The dashed line represents equal performance of both algorithms.

with varying $\overline{\lambda}/\lambda_{\max}$ and axis alignment. We set the eigenvalues as $\mathbf{\Lambda} = \mathrm{diag}(1, 1, \ldots, 1, \lambda_{\max})$. To control the axis-alignment, we rotate the eigenvectors by some ratio $\theta$ in the direction prescribed by a randomly-drawn rotation matrix. We can simply think of $\theta$ as a degree of non-axis-alignment; the technical details can be found in Appendix D. For each Hessian, we compute the two smoothness constants $L_2 = \lambda_{\max}$ and $L_\infty = \|\mathbf{H}\|_{\infty,1}$. We then run $T = 100$ iterations of gradient descent (with $\alpha = 1/L_2$) and sign gradient descent (with $\alpha = 1/L_\infty$) and compute the distance to the optimum $\|\mathbf{x}_T - \mathbf{x}^*\|_2^2$ as a scale-invariant performance measure. We average over repeated runs with $\mathbf{x}_0 \sim \mathcal{N}(0, I)$ to marginalize out the effect of initialization. The results are depicted in Fig. 2 and confirm the findings of Sections 3 and 4. The $L_\infty$ constant—and consequently the performance of signGD—is sensitive to the axis-alignment of $\mathbf{H}$ and suffers as we increase $\theta$. For problems with $\lambda_{\max} \gg \overline{\lambda}$ that are somewhat axis-aligned, sign gradient descent outperforms gradient descent, even on these simple quadratic problems.

## 5.2 NEURAL NETWORK EXPERIMENTS

Variants of sign gradient descent have been shown to achieve surprisingly good performance in neural network training tasks (Balles & Hennig, 2018; Bernstein et al., 2018). In this section, we want to test whether the theory developed above can help explain these successes. Applying our theory to a deep learning scenario comes with several caveats. First, the objectives are *not* globally smooth in the sense of Eq. (4); see also our discussion in Appendix C. However, the theory of smooth optimization still applies in a local sense within a neighborhood of the current iterate. Second, even computing local smoothness constants is intractable in these complex and high-dimensional problems. Finally, neural networks are usually trained in a stochastic optimization setting, whereas our theory considered only the deterministic case.

With these limitations in mind, we proceed as follows. We train a vanilla convolutional neural network as well as a ResNet20 (He et al., 2016) on the CIFAR-10 dataset using gradient descent and sign gradient descent (in their stochastic versions). For each architecture and algorithm, we optimize the step size over a logarithmic grid to make sure we are close to the optimal step size, as our theory assumes. At regular intervals (once per epoch) along the trajectory of the two methods, we approximate the local improvement ratio $\mathcal{R}(\mathbf{x})$ defined in Eq. (12). While the gradient density is easy to compute by evaluating the full-batch gradient, we approximate the two smoothness constants with directional finite differences following Zhang et al. (2019). Details can be found in Appendix D.

We then compare the sequence of ratios to the actual training loss performance of the two methods. We see in Figure 3 that the results match our bound: for the CNN, the ratios are close to 1 but initially bigger and signSGD outperforms SGD. For the ResNet20, the ratio is consistently below 1 and SGD

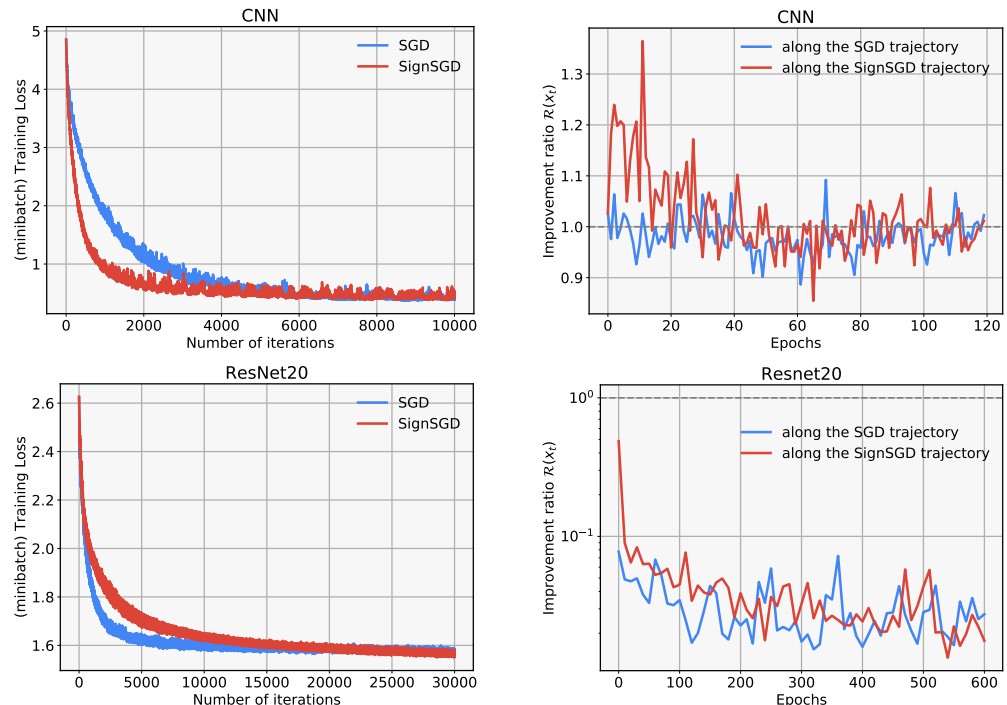

Figure 3: Experiments on CIFAR10 dataset. When using a CNN (top), signSGD outperforms SGD (top left), which matches the behaviour predicted by the improvement ratio (top right), above 1 at the beginning. When using a ResNet20 (bottom), the situation is reversed but the improvement ratio still correctly predicts the relative performance of both algorithms.

is the faster algorithm. We can also note that the ratios computed along the two trajectories do not differ substantially, suggesting some degree of robustness. There are still some issues. For instance, in the second experiment, SignSGD outperforms SGD later in training, which is not captured by our ratio. While these experiments are by no means conclusive, they give a strong indication that the theory of this paper is relevant to explaining the practical performance of sign-based optimization methods.

## 6 RELATED WORK

### 6.1 STEEPEST DESCENT

Steepest descent is a basic concept and has become text book knowledge (Boyd & Vandenberghe, 2004, §9.4). Its origins are hard to trace back. It features—sometimes implicitly—in many works (Nesterov, 2005; 2012; Allen-Zhu & Orecchia, 2014; Kelner et al., 2014; Carlson et al., 2015; Karimi et al., 2016, e.g.). Kelner et al. (2014) and Carlson et al. (2015) point out that sign gradient descent is steepest descent w.r.t. the maximum norm; the former applies signGD to max-flow problems in graphs.

Steepest descent resembles mirror descent (Nemirovsky & Yudin, 1983; Beck & Teboulle, 2003), which updates $\boldsymbol{x}_{t+1} = \arg\min_{\boldsymbol{x}}(\langle \nabla f_t, \boldsymbol{x} - \boldsymbol{x}_t \rangle + \frac{1}{\alpha_t} B_\psi(\boldsymbol{x}, \boldsymbol{x}_t))$, where $B_\psi$ is a Bregman divergence. Despite the obvious similarity, this setting does not cover steepest descent, since many squared norms (notably the maximum norm) can not be written as Bregman divergences.

### 6.2 PREVIOUS WORKS ON SIGN GRADIENT DESCENT

A version of sign gradient descent has first been explored in neural network training by Seide et al. (2014) with the goal of gradient compression for distributed optimization. The proposed algorithm

deviates from signGD in that it uses an error feedback mechanism to approximate gradient descent as closely as possible while maintaining high compression rates. Such error feedback was later refined and given theoretical grounding by Karimireddy et al. (2019).

The use of sign gradient descent itself (or rather its stochastic version) was popularized by Bernstein et al. (2018), who present a theoretical analysis in the stochastic setting. The relatively strong assumptions of this work have since been relaxed by Bernstein et al. (2019) and Safaryan & Richtárik (2019). All three papers are based on the assumption that there exist constants $l_i > 0$ such that

$$f(\boldsymbol{y}) \leq f(\boldsymbol{x}) + \langle \nabla f(\boldsymbol{x}), \boldsymbol{y} - \boldsymbol{x} \rangle + \frac{1}{2} \sum_i l_i (\boldsymbol{y}_i - \boldsymbol{x}_i)^2, \quad \forall \boldsymbol{x}, \boldsymbol{y} \in \mathbb{R}^d . \tag{14}$$

We refer to this assumption as *separable* smoothness to emphasize the fact that the change in function value *separates* over individual coordinates. Under this assumption the convergence of signGD depends on $\sum_i l_i$ whereas the convergence speed of gradient descent is driven by $l_{\max} := \max_i l_i$.

Not only is the meaning of these $l_i$'s unclear, but neither the algorithm itself nor the analysis uses the individual $l_i$'s. It thus seems that the separable smoothness assumption adds an unnecessary level of granularity. Indeed, we will now show that

(i) separable smoothness implies $(\sum_i l_i)$-smoothness with respect to the maximum norm, and

(ii) existing convergence proofs go through under the latter (weaker) assumption.

To formalize (i), we identify separable smoothness as 1-smoothness w.r.t. the norm $\| \cdot \|_{\boldsymbol{L}}$ where $\boldsymbol{L} = \mathrm{diag}(l_1, \ldots, l_d$ and $\|\boldsymbol{z}\|_{\boldsymbol{L}}^2 := \left( \sum_i l_i z_i^2 \right)$, such that Lemma 1 coincides with Eq. (14). With that, we can establish the following result:

**Proposition 4.** *If $f$ is 1-smooth w.r.t. $\| \cdot \|_{\boldsymbol{L}}$, then $f$ is $(\sum_i l_i)$-smooth w.r.t. the maximum norm.*

Statement (ii) follows from the simple fact that the quadratic bounds provided by the two conditions *coincide* for any method with updates of equal magnitude in each coordinate, $\boldsymbol{x}_{t+1} - \boldsymbol{x}_t \in \{-c, c\}^d$, which includes any version of sign gradient descent. Since existing convergence results base upon this bound, they go through under either condition. We refer to Appendix B for details.

Based on these considerations we argue that smoothness w.r.t. the maximum norm is the more natural assumption under which to analyze sign gradient descent. We discuss this issue in much more detail in Appendix B.

## 7 CONCLUSIONS

Our analysis relates the speed of signGD to the axis-alignment of the objective's Hessian and the spread of its eigenvalues. This furthers our understanding of when this method and similar ones, like Adam, might outperform standard GD. It is notably of interest that these properties of the Hessian have independently been studied (Chaudhari et al., 2017; Ghorbani et al., 2019; Adolphs et al., 2019) with the conclusion that modern architecture tend to be well-suited to the use of signGD.

Besides the limitations discussed in §5.2, we also sidestepped the question of the scaling of the sign gradient descent update with $\|\nabla f_t\|_1$, which arises from the steepest descent formalism but is not common in practice. Appendix C discusses this issue and how the absence of the scaling term makes sense in a more general setting than the one studied here.

This work raises several questions for future research. The first one is whether we can design architecture that could further help the convergence of signGD. The second one is whether we can design optimizers that adapt to the geometry of the space. Ideally, one might want the norm of steepest descent to be optimized alongside the parameters. Finally, one might ask whether there are other norms and corresponding steepest descent methods that work even better on contemporary neural network architectures. We explore one option, block-normalized gradient descent, in Appendix A.

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

—SUPPLEMENTARY MATERIAL—

We now provide additional details and results for the paper. More precisely:

- Appendix A gives additional details on steepest descent methods.
- Appendix B extends the discussion on the relationship between separable smoothness and smoothness w.r.t. the maximum norm.
- Appendix C discusses an extension of a recently-proposed relaxed smoothness condition as a possible explanation of normalized methods (such as sign gradient descent without the scaling by the norm).
- Appendix D provides details on the experiments.
- All proofs, including those of results in the appendices, can be found in Appendix E.

## A DETAILS ON STEEPEST DESCENT

In this section, we provide a more comprehensive overview of steepest descent methods as well as additional results. In particular, we recall the linear convergence under the Polyar-Łojasiewicz condition.

It will be useful to re-write the steepest descent update (Eq. 2) as

$$\boldsymbol{x}_{t+1} = \boldsymbol{x}_t - \alpha_t \nabla f_t^{\|\cdot\|} \quad \text{with} \quad \boldsymbol{z}^{\|\cdot\|} \in \arg\max_{\boldsymbol{y} \in \mathbb{R}^d} \left( \langle \boldsymbol{z}, \boldsymbol{y} \rangle - \frac{1}{2} \|\boldsymbol{y}\|^2 \right) . \tag{15}$$

This equivalence arises from substituting $\boldsymbol{y} = -\frac{1}{\alpha_t}(\boldsymbol{x} - \boldsymbol{x}_t)$; a full proof can be found in Appendix E

### A.1 ADDITIONAL EXAMPLES FOR STEEPEST DESCENT METHODS

First, we give two additional examples for steepest descent methods to demonstrate the versatility of this framework.

#### A.1.1 COORDINATE DESCENT

Steepest descent with respect to the $L^1$-norm,

$$\boldsymbol{x}_{t+1} \in \arg\min \left( f_t + \langle \nabla f_t, \boldsymbol{x} - \boldsymbol{x}_t \rangle + \frac{1}{2\alpha_t} \|\boldsymbol{x} - \boldsymbol{x}_t\|_1^2 \right), \tag{16}$$

yields coordinate descent,

$$\boldsymbol{x}_{t+1} = \boldsymbol{x}_t - \alpha_t |\nabla f_{t,i_{\max}}| e^{(i_{\max})} \tag{17}$$

where the selected coordinate is chosen from $i_{\max} \in \arg\max_{i \in [d]} |\nabla f_{t,i}|$ and $\boldsymbol{e}^{(i)}$ denotes the $i$-th coordinate vector. The corresponding smoothness assumption is

$$\|\nabla f(\boldsymbol{y}) - \nabla f(\boldsymbol{x})\|_\infty \le L_1 \|\boldsymbol{y} - \boldsymbol{x}\|_1. \tag{18}$$

In fact, this assumption implies the coordinate-wise Lipschitz smoothness assumption,

$$|\nabla f(\boldsymbol{x} + h\boldsymbol{e}^{(i)})_i - \nabla f(\boldsymbol{x})_i| \le L_1 |h| \quad \forall i \in [d], \tag{19}$$

which is widely-used in the literature on coordinate descent, since

$$|\nabla f(\boldsymbol{x} + h\boldsymbol{e}^{(i)})_i - \nabla f(\boldsymbol{x})_i| \le \|\nabla f(\boldsymbol{x} + h\boldsymbol{e}^{(i)}) - \nabla f(\boldsymbol{x})\|_\infty$$
$$\overset{(18)}{\le} L_1 \|\boldsymbol{x} + h\boldsymbol{e}^{(i)} - \boldsymbol{x}\|_1 = L_1 |h|. \tag{20}$$

### A.1.2 BLOCK-NORMALIZED GRADIENT DESCENT

Assume a block structure on $\mathbb{R}^d$ given by a partitioning $\mathcal{B} = \{B_1, \ldots, B_b\}$ of $[d]$, with $B_k \subset [d]$, $B_k \cap B_l = \emptyset$ for $k \neq l$, and $\bigcup_k B_k = [d]$. For $B \subset [d]$, define $\boldsymbol{x}_B \in \mathbb{R}^{|B|}$ to be the vector consisting of $(x_i)_{i \in B}$. We can now define norms with respect to this block structure, such as

$$\|\boldsymbol{x}\|_{\mathcal{B},\infty} = \max_{B \in \mathcal{B}} \|\boldsymbol{x}_B\|_2 \quad \text{with dual norm} \quad \|\boldsymbol{x}\|_{\mathcal{B},1} = \sum_{B \in \mathcal{B}} \|\boldsymbol{x}_B\|_2. \tag{21}$$

Steepest descent w.r.t. $\|\cdot\|_{\mathcal{B},\infty}$ results in *block-normalized gradient descent*,

$$\nabla f^{\|\cdot\|_{\mathcal{B},\infty}} = \|\nabla f\|_{\mathcal{B},1} \operatorname{norm}_{\mathcal{B}}(\nabla f), \quad \operatorname{norm}_{\mathcal{B}}(\boldsymbol{z}) = \left( \frac{\boldsymbol{z}_{B_1}^T}{\|\boldsymbol{z}_{B_1}\|_2}, \ldots, \frac{\boldsymbol{z}_{B_b}^T}{\|\boldsymbol{z}_{B_b}\|_2} \right)^T. \tag{22}$$

This method is a block-wise equivalent of sign gradient descent, normalizing the update magnitude over blocks instead of element-wise. Variants of this method have recently been studied empirically for neural network training (Yu et al., 2017; Ginsburg et al., 2019). It would be interesting future work to analyze this method in the steepest descent framework, similar to our analysis of sign gradient descent in this paper.

### A.2 PROPERTIES OF THE STEEPEST DESCENT OPERATOR

We define the $\boldsymbol{z}^{\|\cdot\|}$ operator as:

$$\|\boldsymbol{z}\|_* := \max_{\|\boldsymbol{x}\| \leq 1} \langle \boldsymbol{z}, \boldsymbol{x} \rangle \quad , \quad \boldsymbol{z}^{\|\cdot\|} \in \arg\max_{\boldsymbol{y} \in \mathbb{R}^d} \left( \langle \boldsymbol{z}, \boldsymbol{y} \rangle - \frac{1}{2} \|\boldsymbol{y}\|^2 \right). \tag{23}$$

The following Lemma connects this operator to the dual norm:

**Lemma 3.** *For all* $\boldsymbol{x}, \boldsymbol{z} \in \mathbb{R}^d$, *we have*

$$\text{(a)} \quad \langle \boldsymbol{x}, \boldsymbol{z} \rangle \leq \|\boldsymbol{x}\| \|\boldsymbol{z}\|_* \tag{24}$$

$$\text{(b)} \quad \|\boldsymbol{z}^{\|\cdot\|}\|^2 = \langle \boldsymbol{z}, \boldsymbol{z}^{\|\cdot\|} \rangle \tag{25}$$

$$\text{(c)} \quad \|\boldsymbol{z}^{\|\cdot\|}\| = \|\boldsymbol{z}\|_* \tag{26}$$

### A.3 ADDITIONAL CONVERGENCE RESULTS FOR STEEPEST DESCENT

For smooth and convex functions, Kelner et al. (2014) showed $O(1/T)$ convergence in suboptimality. We restate this result here for completeness.

**Theorem 1** (Theorem 1 in Kelner et al. (2014))**.** *If $f$ is $L$-smooth w.r.t.* $\|\cdot\|$ *and convex, then steepest descent (Eq. 2) with step size* $\alpha_t = 1/L$ *satisfies*

$$f_T - f^\star \leq \frac{2LR^2}{T+4} \tag{27}$$

$$\text{with } R := \max_{\boldsymbol{x} \text{ s.t. } f(\boldsymbol{x}) \leq f(\boldsymbol{x}_0)} \min_{\boldsymbol{x}^\star \text{ s.t. } f(\boldsymbol{x}^\star) = f^\star} \|\boldsymbol{x} - \boldsymbol{x}^\star\|.$$

The rate has a dependence on the initial distance to the nearest minimizer measured in the respective norm.

It is also straight-forward to show linear convergence in suboptimality under an additional assumption, known as the Polyak-Łojasiewicz (PL) condition. It is usually given for the Euclidean case as $\|\nabla f(\boldsymbol{x})\|_2^2 \geq 2\mu(f(\boldsymbol{x}) - f^\star)$ but can likewise be formulated for arbitrary norms.

**Definition 1.** *A function* $f: \mathbb{R}^d \to \mathbb{R}$ *satisfies the PL condition w.r.t. a norm* $\|\cdot\|$ *if* $\|\nabla f(\boldsymbol{x})\|_*^2 \geq 2\mu(f(\boldsymbol{x}) - f^\star)$ *for all* $\boldsymbol{x} \in \mathbb{R}^d$.

We refer to this as PL with respect to $\|\cdot\|$ even though only the dual norm appears in the definition, since it is the natural counterpart to smoothness w.r.t. $\|\cdot\|$ and used to prove linear convergence for steepest descent w.r.t. $\|\cdot\|$. As with smoothness, we have equivalence of the PL condition for all norms, but constants may differ. Note that strong convexity implies the PL condition, but the class of PL functions also covers some non-convex functions.

**Theorem 2.** *If $f$ is L-smooth and fulfills the PL condition with constant $\mu$ w.r.t $\| \cdot \|$, then steepest descent (Eq. 2) with step size $\alpha_t \equiv 1/L$ satisfies*

$$f_T - f^\star \leq \left(1 - \frac{\mu}{L}\right)^T (f_0 - f^\star). \tag{28}$$

We are not aware of any published work showing this simple result in its general form. We give the very short proof here.

*Proof of Theorem 2.* Combining Lemma 2 and the PL condition gives

$$f_{t+1} \leq f_t - \frac{1}{2L}\|\nabla f_t\|_*^2 \tag{29}$$

$$\leq f_t - \frac{\mu}{L}(f_t - f^\star). \tag{30}$$

Subtracting $f^\star$ from both sides and iterating backwards yields the statement. $\square$

# B    DETAILS ON THE RELATIONSHIP BETWEEN SEPARABLE SMOOTHNESS AND SMOOTHNESS W.R.T. THE MAXIMUM NORM

This section presents additional insights on the relationship between separable smoothness and smoothness w.r.t. the maximum norm. We discuss how separable smoothness relates to properties of the Hessian of the objective and give a two-dimensional quadratic example to support intuition. Finally, we show that existing convergence proofs for (stochastic) sign gradient descent go through when replacing the separable smoothness assumption with the weaker smoothness w.r.t. the maximum norm.

## B.1    SEPARABLE SMOOTHNESS AS A DIAGONAL BOUND ON THE HESSIAN

Bernstein et al. (2018) note that, for twice-differentiable $f$, separable smoothness results from $-\boldsymbol{L} \preceq \nabla^2 f(\boldsymbol{x}) \preceq \boldsymbol{L}$ for all $\boldsymbol{x} \in \mathbb{R}^d$ with $\boldsymbol{L} = \mathrm{diag}(l_1, \ldots, l_d)$. It is tempting to think of them as bounds on the eigenvalues of $\nabla^2 f(\boldsymbol{x})$, but this is only true for diagonal matrices. In the following, we will see that these values also depend on the axis-alignment of the Hessian and will generally exceed the eigenvalues. Moreover, since "$\preceq$" is only a partial ordering, it is not clear what the "tightest" possible bound of this form would be. Since the performance of signGD depends on $\sum_i l_i$, the most favorable bound will be

$$\min_{l_1,\ldots,l_d \geq 0} \sum_i l_i \quad \text{s.t.} \quad -\boldsymbol{L} \preceq \nabla^2 f(\boldsymbol{x}) \preceq \boldsymbol{L} \ \forall \boldsymbol{x} \in \mathbb{R}^d \tag{31}$$

We will adopt this definition in the following.

This opens up an alternative view on the relationship between separable smoothness and smoothness w.r.t. the maximum norm, which operates on the Hessian-based definitions of the two notions of smoothness.

**Proposition 5.** *Let $\boldsymbol{H} \in \mathbb{R}^{d \times d}$ be positive semi-definite with eigenvalues $\lambda_1 \ldots, \lambda_d$ and define*

$$L_\infty(\boldsymbol{H}) := \max_{\|\boldsymbol{x}\|_\infty \leq 1} \|\boldsymbol{H}\boldsymbol{x}\|_1, \qquad L_{sep}(\boldsymbol{H}) := \min_{l_i \geq 0} \sum_i l_i \quad s.t. \quad \boldsymbol{H} \preceq \mathrm{diag}(l_1, \ldots, l_d). \tag{32}$$

*Then*

$$L_\infty(\boldsymbol{H}) \leq L_{sep}(\boldsymbol{H}) \leq \rho_{diag}(\boldsymbol{H})^{-1} \sum_i \lambda_i. \tag{33}$$

Hence, $L_{\text{sep}}$ upper-bounds $L_\infty$ while at the same time being upper-bounded by the same quantity that appears in Proposition 3.

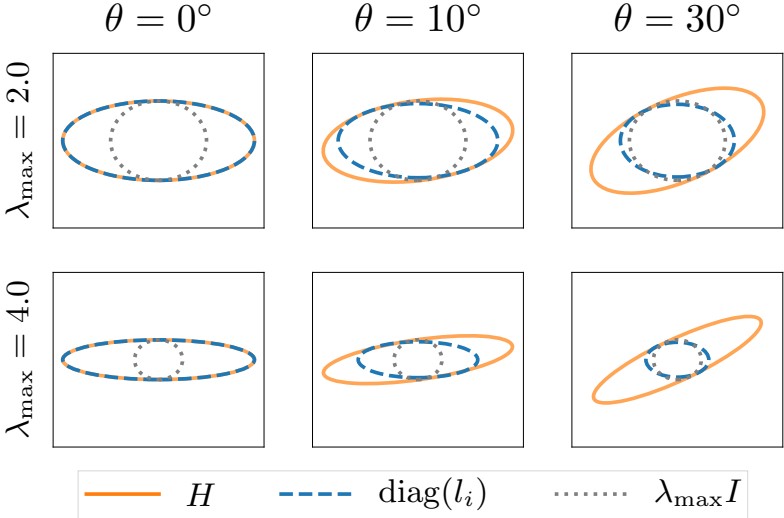

Figure 4: For $\boldsymbol{H} \in \mathbb{R}^{d \times d}$, we plot a contour line of $f(\boldsymbol{x}) = \frac{1}{2}\boldsymbol{x}^T\boldsymbol{H}\boldsymbol{x}$, which forms an ellipse with principal axes given by the eigenvectors and axis lengths given by the inverse eigenvalues. We fix $\lambda_{\min} = 1$ and vary $\lambda_{\max} > 1$ as well as the angle $\theta$ between eigenvectors and coordinate axes. Separable smoothness bounds $\boldsymbol{H}$ by a diagonal matrix, $\mathrm{diag}(l_i) \succeq \boldsymbol{H}$, corresponding to an axis-aligned ellipse that lies fully within the $\boldsymbol{H}$-ellipse. (We choose $l_i$ according to Eq. 31.) The bound changes with the axis-alignment of $\boldsymbol{H}$, becoming both smaller and more circular (i.e., larger and more similar $l_i$) as $\boldsymbol{H}$ rotates further from the coordinate axes. In contrast to that, Euclidean smoothness bounds $\boldsymbol{H}$ by $\lambda_{\max}\boldsymbol{I} \succeq \boldsymbol{H}$, i.e., a circle, which is rotation-invariant.

### B.2 SEPARABLE SMOOTHNESS FOR TWO-DIMENSIONAL QUADRATICS

To build some intuition for the separable smoothness condition, we consider the case of a two-dimensional quadratic with positive definite Hessian $\nabla^2 f(\boldsymbol{x}) \equiv \boldsymbol{H} = \left[ \begin{smallmatrix} a & b \\ b & d \end{smallmatrix} \right]$. In this, case, we can find the solution to Eq. (31) in closed form (proof in Appendix E):

$$l_1 = a + |b|, \quad l_2 = d + |b|. \tag{34}$$

A visualization is given in Figure 4 and clearly shows that the choice of $l_i$ is sensitive to the axis alignment of the Hessian.

The eigenvalues of $\boldsymbol{H}$ evaluate to

$$\lambda_{1/2} = \frac{a+d}{2} \pm \sqrt{\frac{(a-d)^2}{4} + b^2}. \tag{35}$$

We see that, for $|b| > 0$, we will have $l_{\max} > \lambda_{\max}$.[4] We also see that $l_{\max}$ can exceed $\lambda_{\max}$ and it is thus misleading to assume the convergence of GD to be driven by $l_{\max}$.

Also note that $L_\infty = \|\boldsymbol{H}\|_{\infty,1} = a + d + 2|b| = \sum_i l_i$, reflecting the link to smoothness w.r.t. the maximum norm, as discussed in the main text.

---

[4]Of course, we could deviate from the definition of Eq. (31) and choose $l_i \equiv \lambda_{\max}$ to guarantee $l_{\max} = \lambda_{\max}$. However, this bound would not be favorable for the performance of signGD, which depends on $\sum_i l_i$.

### B.3 EXISTING CONVERGENCE PROOFS GO TROUGH UNDER SMOOTHNESS W.R.T. THE MAXIMUM NORM

Let $\boldsymbol{x}_{t+1} - \boldsymbol{x}_t \in \{-c, c\}^d$ for some $c \in \mathbb{R}$. Then the separable smoothness bound implies

$$
\begin{aligned}
f_{t+1} &\leq f_t + \langle \nabla f_t, \boldsymbol{x}_{t+1} - \boldsymbol{x}_t \rangle + \frac{1}{2} \sum_i l_i \underbrace{(x_{t+1,i} - x_{t,i})^2}_{=c^2} \\
&= f_t + \langle \nabla f_t, \boldsymbol{x}_{t+1} - \boldsymbol{x}_t \rangle + \frac{c^2}{2} \sum_i l_i
\end{aligned}
\tag{36}
$$

and the $L_\infty$-smoothness w.r.t. the maximum norm yields (via Lemma 1)

$$
\begin{aligned}
f_{t+1} &\leq f_t + \langle \nabla f_t, \boldsymbol{x}_{t+1} - \boldsymbol{x}_t \rangle + \frac{1}{2} L_\infty \underbrace{\|\boldsymbol{x}_{t+1} - \boldsymbol{x}_t\|_\infty^2}_{=c^2} \\
&= f_t + \langle \nabla f_t, \boldsymbol{x}_{t+1} - \boldsymbol{x}_t \rangle + \frac{c^2}{2} L_\infty.
\end{aligned}
\tag{37}
$$

Hence, smoothness w.r.t. the maximum norm with constant $L_\infty = \sum_i l_i$ provides exactly the same local bound for any method with updates of equal magnitude in each coordinate. This includes any version of sign gradient descent, including in the stochastic setting. Since the convergence proofs given by Bernstein et al. (2018; 2019) and Safaryan & Richtárik (2019) all start from this bound, they go through when replacing the separable smoothness assumption with the weaker assumption of $(\sum_i l_i)$-smoothness w.r.t. the maximum norm.

## C    GENERALIZING RELAXED SMOOTHNESS

We have discussed a scaled version of sign gradient descent, $\boldsymbol{x}_{t+1} = \boldsymbol{x}_t - \alpha \|\nabla f_t\|_1 \operatorname{sign}(\nabla f_t)$, whereas in practice the update is usually $\boldsymbol{x}_{t+1} = \boldsymbol{x}_t - \alpha_t \operatorname{sign}(\nabla f_t)$ with a constant or manually decreasing step size $\alpha_t$. Without this additional $\|\nabla f_t\|_1$ term, this is a *normalized* method with an update magnitude determined solely by the step size, independent of the gradient $\nabla f_t$. There are many possible reasons why this might be beneficial in practical settings like neural network training. One rationale is that these applications are usually in the stochastic optimization setting, where a decreasing step size is needed anyway to enforce convergence. Since $\|\nabla f_t\|_1$ is not available in the stochastic setting, it might be easier to subsume a similar scaling effect in the manually-tuned step size schedule.

There is however another potential explanation in terms of the smoothness exhibited by neural network training objectives. Zhang et al. (2019) discussed the normalized gradient descent method,

$$
\boldsymbol{x}_{t+1} = \boldsymbol{x}_t - \alpha \frac{1}{\|\nabla f_t\|_2 + \beta} \nabla f_t.
\tag{38}
$$

This method is *normalized* since its update magnitude is upper-bounded by $\|\boldsymbol{x}_{t+1} - \boldsymbol{x}_t\|_2 \leq \alpha$ irrespective of the gradient magnitude. For small gradient $\|\nabla f_t\|_2 \ll \beta$, the method reverts back to classical gradient descent. They show that this normalized method is better geared towards the type of regularity exhibited by, say, neural network training objectives, which are not smooth in the sense of Eq. (4). They propose a "relaxed" smoothness assumption which allows the curvature to grow with the gradient norm instead of bounding it globally as in classical smoothness.

Since they consider normalized gradient descent, their discussion is based on Euclidean geometry. We show here that their reasoning can be generalized to arbitrary norms, similarly to the standard smoothness assumption discussed in the main text. This provides a possible explanation of the practical success of *normalized* methods, which sign gradient descent (Eq. 1) is if we omit the scaling by $\|\nabla f_t\|_1$.

### C.1    RESULTS OF ZHANG ET AL. (2019)

The relaxed smoothness condition proposed by Zhang et al. (2019) reads

$$
\|\nabla^2 f(\boldsymbol{x})\|_2 \leq L^{(0)} + L^{(1)} \|\nabla f(\boldsymbol{x})\|_2,
\tag{39}
$$

where $\|\cdot\|_2$ for matrices denotes the spectral norm. This allows for the curvature to grow with the gradient norm, in contrast to classical smoothness, which demands a global bound on the Hessian.

This relaxed smoothness gives rise to normalized gradient descent since, as we will see later, provides a bound of the form

$$f_{t+1} \leq f_t + \langle \nabla f_t, \boldsymbol{x}_{t+1} - \boldsymbol{x}_t \rangle + \frac{1}{2}(A + B\|\nabla f_t\|_2)\|\boldsymbol{x}_{t+1} - \boldsymbol{x}_t\|_2^2, \quad A, B \geq 0. \tag{40}$$

This resembles the bound of Lemma 1, but the quadratic term now scales with the gradient norm. It is minimized by a normalized gradient descent update with appropriately chosen $\alpha$ and $\beta$.

The main finding of Zhang et al. (2019) is that gradient descent can become arbitrarily slow for the class of functions satisfying this relaxed smoothness, whereas normalized gradient descent (Eq. 38) retains an $O(1/\varepsilon^2)$ rate of convergence to a $\varepsilon$-stationary point.

## C.2 NORMALIZED STEEPEST DESCENT

In this section, we generalize the concept of relaxed smoothness to arbitrary norms, which will give rise to general normalized steepest descent methods. We define relaxed smoothness w.r.t. to some norm $\|\cdot\|$ analogously to the Euclidean case (Eq. 39), but use the dual norm for the gradient and the induced matrix norm of Proposition 2 for the Hessian.

**Definition 2.** *A function $f$ is called $(L^{(0)}, L^{(1)})$-smooth with respect to some norm $\|\cdot\|$ if*

$$\|\nabla^2 f(\boldsymbol{x})\| \leq L^{(0)} + L^{(1)}\|\nabla f(\boldsymbol{x})\|_*, \tag{41}$$

*where $\|\cdot\|$ for matrices is the norm defined in Eq. (8).*

Under this smoothness assumption, we have the following local quadratic bound:

**Lemma 4.** *Assume $f$ is $(L^{(0)}, L^{(1)})$-smooth with respect to a norm $\|\cdot\|$. Then for $\boldsymbol{x}, \boldsymbol{y} \in \mathbb{R}^d$ with $\|\boldsymbol{y} - \boldsymbol{x}\| \leq \frac{1}{L^{(1)}}$,*

$$f(\boldsymbol{y}) \leq f(\boldsymbol{x}) + \langle \nabla f(\boldsymbol{x}), \boldsymbol{y} - \boldsymbol{x} \rangle + \frac{1}{2}(5L^{(0)} + 4L^{(1)}\|\nabla f(\boldsymbol{x})\|_*)\|\boldsymbol{y} - \boldsymbol{x}\|^2. \tag{42}$$

This resembles the bound in Lemma 1, but the quadratic term now scales with the gradient norm. In analogy to steepest descent, we can now construct an optimization method that minimizes this bound in each step. Using Lemma 4 with $\boldsymbol{x} = \boldsymbol{x}_t$ yields

$$f(\boldsymbol{y}) \leq f_t + \langle \nabla f_t, \boldsymbol{y} - \boldsymbol{x}_t \rangle + \frac{1}{2}(5L^{(0)} + 4L^{(1)}\|\nabla f_t\|_*)\|\boldsymbol{y} - \boldsymbol{x}_t\|^2. \tag{43}$$

Choosing $\boldsymbol{x}_{t+1}$ as the minimizer w.r.t. $\boldsymbol{y}$ results in the update

$$\boldsymbol{x}_{t+1} = \boldsymbol{x}_t - \frac{1}{(5L^{(0)} + 4L^{(1)}\|\nabla f_t\|_*)}\nabla f_t^{\|\cdot\|}. \tag{44}$$

We refer to the resulting method as normalized steepest descent. To see that it is in fact a *normalized* method, recall from Lemma 3 that $\|\nabla f_t^{\|\cdot\|}\| = \|\nabla f_t\|_*$ and, hence, the update magnitude is upper-bounded by $1/4L^{(1)}$.

We now show that the convergence theorem of Zhang et al. (2019) carries over to this generalized setting. The proofs (see Appendix E) are straight-forward adaptations of that in Zhang et al. (2019), with a little bit of extra care with regards to the norms.

**Theorem 3.** *Assume $f$ is $(L^{(0)}, L^{(1)})$-smooth with respect to a norm $\|\cdot\|$. Then normalized steepest descent (Eq. 44) converges to an $\varepsilon$-stationary point, $\|\nabla f\|_* \leq \varepsilon$, in at most*

$$T_\varepsilon = 18(f_0 - f_*)\max\left(\frac{L^{(0)}}{\varepsilon^2}, \frac{(L^{(1)})^2}{L^{(0)}}\right) \tag{45}$$

*iterations.*

## D    EXPERIMENTAL DETAILS

### D.1    QUADRATIC EXPERIMENTS

**Generating Hessians.**    W draw a random rotation matrix $\boldsymbol{R}$ from the Haar distribution[5] and set the Hessian to be $\boldsymbol{H} = \boldsymbol{R}^\theta \boldsymbol{\Lambda} (\boldsymbol{R}^\theta)^*$, where $\boldsymbol{R}^\theta$ for $\theta \in [0, 1]$ is a non-integer matrix power and $\boldsymbol{A}^*$ denotes the conjugate transpose matrix. We can think of this as rotating the eigenvectors of the Hessian by a fraction of $\theta$ in the direction prescribed by $\boldsymbol{R}$. The non-integer matrix power $\boldsymbol{R}^\theta$, is computed via the eigendecomposition $\boldsymbol{R} = \boldsymbol{U}\boldsymbol{D}\boldsymbol{U}^*$. The matrix power is then given by $\boldsymbol{R}^\theta = \boldsymbol{U}\boldsymbol{D}^\theta \boldsymbol{U}^H$ where $\boldsymbol{D}^\theta$ for the diagonal matrix $\boldsymbol{D}$ is obtained by its elements to the power $\theta$.

**Computing $L_\infty$.**    To compute the smoothness constant w.r.t. the maximum norm, we have to compute the matrix norm $\|\boldsymbol{H}\|_{\infty,1} = \max_{\|\boldsymbol{x}\|_\infty \leq 1} \|\boldsymbol{H}\boldsymbol{x}\|_1$. We use the fact that the solution is attained at $\boldsymbol{x} \in \{-1, 1\}^d$ (see Rohn, 2000) and brute-force search for the maximum $\|\boldsymbol{H}\boldsymbol{x}\|_1$ in this set. Since there are $2^d$ vectors in $\{-1, 1\}^d$, this is only possible for relatively small dimension.

**On the performance measure.**    When comparing gradient descent and sign gradient descent on these quadratic problems, we use the distance to the optimum as a performance measure. The reason is that we are interested in a comparison over a range of different quadratics with varying $\lambda_{\max}$. The function value, which scales with $\lambda_{\max}$ would not be suitable for such a comparison. Since we are comparing optimization methods which are adapted to different norms, it might make a difference which norm we choose to compute the distance to the optimum. We opted for the Euclidean norm to benefit the baseline method (gradient descent) as the lesser of two evils.

### D.2    NEURAL NETWORK EXPERIMENTS

In this section we give mode details on the experimental setting used in Figure 3:

**Dataset.**    We used the CIFAR10 (Krizhevsky et al., 2009) dataset, which contains 50,000 images of size (32, 32, 3). In all cases we used a batch-size of 1000 samples.

**Architecture.**    We considered two different architectures for these experiments. The first one is a convolutional neural network (CNN) as implemented in keras,[6] consisting of 4 convolutional layers with ReLU activations, and one last dense layer with a softmax activation function. The second architecture that we consider is a ResNet20, (He et al., 2016) model, consisting of 20 layers and 0.27 million parameters.

**Algorithm and hyperparameter selection.**    We consider both SGD and signSGD with an added momentum term since this matches most closely the algorithms used in practice for this task. For SignSGD, the sign is computed in this case on the full update, and not only on the gradient.

These algorithms have two hyperparameters to tune: the step-size and momentum.

- We select the step-size independently for each algorithm and architecture to yield the highest decrease in training loss at 200 and 600 epochs for the CNN and ResNet20 architecture, respectively, which is when both architectures reach 85% test set accuracy.

- For the momentum parameter, we use the Keras default of 0.9 for all optimizers and architectures.

---

[5]The uniform distribution on the special orthogonal group $SO(d)$ of $d$-dimensional rotation matrices. We used the `special_ortho_group` function provided by the `scipy.stats` package (Jones et al., 2001).

[6]`https://github.com/keras-team/keras/blob/master/examples/cifar10_cnn.py`

**Estimating smoothness constants.** Following Zhang et al. (2019), we approximate the smoothness using directional finite differences. At a point $\boldsymbol{x}$, we approximate

$$\hat{L}_2 = \min_{\gamma} \frac{\|\nabla f(\boldsymbol{x} + \gamma \boldsymbol{s}) - \nabla f(\boldsymbol{x})\|_2}{\gamma \|\boldsymbol{s}\|_2} \quad .\boldsymbol{s} = -\alpha_1 \nabla f(\boldsymbol{x}) \tag{46}$$

$$\hat{L}_\infty = \min_{\gamma} \frac{\|\nabla f(\boldsymbol{x} + \gamma \boldsymbol{s}) - \nabla f(\boldsymbol{x})\|_1}{\gamma \|\boldsymbol{s}\|_\infty} \quad .\boldsymbol{s} = -\alpha_2 \operatorname{sign}(\nabla f(\boldsymbol{x})), \tag{47}$$

where $\alpha_1$ and $\alpha_2$ are the respective "optimal" step size for GD and signGD.

# E    PROOFS

We now list the proofs of all theoretical results from the main text.

## E.1    PROOF OF LEMMA 1

*Proof.* Define $g(\tau) = f(\boldsymbol{x} + \tau(\boldsymbol{y} - \boldsymbol{x}))$ for $\tau \in [0, 1]$ with $g'(\tau) = \langle \nabla f(\boldsymbol{x} + \tau(\boldsymbol{y} - \boldsymbol{x})), \boldsymbol{y} - \boldsymbol{x} \rangle$. Then

$$
\begin{aligned}
f(\boldsymbol{y}) &= f(\boldsymbol{x}) + \int_0^1 g'(\tau) d\tau \\
&= f(\boldsymbol{x}) + \int_0^1 \langle \nabla f(\boldsymbol{x} + \tau(\boldsymbol{y} - \boldsymbol{x})), \boldsymbol{y} - \boldsymbol{x} \rangle d\tau \\
&= f(\boldsymbol{x}) + \langle \nabla f(\boldsymbol{x}), \boldsymbol{y} - \boldsymbol{x} \rangle + \int_0^1 \langle \nabla f(\boldsymbol{x} + \tau(\boldsymbol{y} - \boldsymbol{x})) - \nabla f(\boldsymbol{x}), \boldsymbol{y} - \boldsymbol{x} \rangle d\tau \\
&\overset{(24)}{\leq} f(\boldsymbol{x}) + \langle \nabla f(\boldsymbol{x}), \boldsymbol{y} - \boldsymbol{x} \rangle + \int_0^1 \|\nabla f(\boldsymbol{x} + \tau(\boldsymbol{y} - \boldsymbol{x})) - \nabla f(\boldsymbol{x})\|_* \|\boldsymbol{y} - \boldsymbol{x}\| d\tau \\
&\overset{(4)}{\leq} f(\boldsymbol{x}) + \langle \nabla f(\boldsymbol{x}), \boldsymbol{y} - \boldsymbol{x} \rangle + \int_0^1 L \|\tau(\boldsymbol{y} - \boldsymbol{x})\| \|\boldsymbol{y} - \boldsymbol{x}\| d\tau \\
&= f(\boldsymbol{x}) + \langle \nabla f(\boldsymbol{x}), \boldsymbol{y} - \boldsymbol{x} \rangle + L \|\boldsymbol{y} - \boldsymbol{x}\|^2 \int_0^1 \tau d\tau \\
&= f(\boldsymbol{x}) + \langle \nabla f(\boldsymbol{x}), \boldsymbol{y} - \boldsymbol{x} \rangle + \frac{L}{2} \|\boldsymbol{y} - \boldsymbol{x}\|^2.
\end{aligned}
\tag{48}
$$

The first inequality is due to Lemma 3(a) and the second inequality uses the $L$-smoothness. $\square$

## E.2    CONVERGENCE PROOFS FOR STEEPEST DESCENT

*Proof of Lemma 2.* We apply Lemma 1 with $\boldsymbol{y} = \boldsymbol{x}^+ = \boldsymbol{x} - \frac{1}{L} \nabla f(\boldsymbol{x})^{\|\cdot\|}$

$$
\begin{aligned}
f(\boldsymbol{x}^+) &\leq f(\boldsymbol{x}) + \langle \nabla f(\boldsymbol{x}), \boldsymbol{x}^+ - \boldsymbol{x} \rangle + \frac{L}{2} \|\boldsymbol{x}^+ - \boldsymbol{x}\|^2 \\
&= f(\boldsymbol{x}) + \langle \nabla f(\boldsymbol{x}), -\frac{1}{L} \nabla f(\boldsymbol{x})^{\|\cdot\|} \rangle + \frac{L}{2} \left\| -\frac{1}{L} \nabla f(\boldsymbol{x})^{\|\cdot\|} \right\|^2 \\
&= f(\boldsymbol{x}) - \frac{1}{L} \left( \langle \nabla f(\boldsymbol{x}), \nabla f(\boldsymbol{x})^{\|\cdot\|} \rangle - \frac{1}{2} \|\nabla f(\boldsymbol{x})^{\|\cdot\|}\|^2 \right).
\end{aligned}
\tag{49}
$$

By Lemma 3, we have $\langle \nabla f(\boldsymbol{x}), \nabla f(\boldsymbol{x})^{\|\cdot\|} \rangle = \|\nabla f(\boldsymbol{x})^{\|\cdot\|}\|^2 = \|\nabla f(\boldsymbol{x})\|_*^2$. Substituting this in yields the desired bound. $\square$

*Proof of Proposition 1.* Lemma 2 gives

$$\|\nabla f_t\|_*^2 \leq 2L(f_t - f_{t+1}) \tag{50}$$

Rearranging and summing for $t = 0, \dots, T-1$ yields

$$\frac{1}{T} \sum_{t=0}^{T-1} \|\nabla f_t\|_*^2 \leq \frac{2L}{T} \sum_{t=0}^{T-1} (f_t - f_{t+1}) = \frac{2L(f_0 - f_T)}{T} \leq \frac{2L(f_0 - f^\star)}{T}. \tag{51}$$

$\square$

### E.3 PROOF OF PROPOSITION 2

Note that the matrix norm by construction satisfies

$$\|\boldsymbol{H}\boldsymbol{x}\|_* \le \|\boldsymbol{H}\|\|\boldsymbol{x}\|. \tag{52}$$

*Proof.* We first show that Eq. (8) defines a matrix norm. Clearly $\|\boldsymbol{H}\| \ge 0$ and $\|\boldsymbol{H}\| = 0$ iff $\boldsymbol{H} = 0$. Furthermore, $\|\lambda \boldsymbol{H}\| = |\lambda|\|\boldsymbol{H}\|$. It remains to show subadditivity. Let $\boldsymbol{H}, \boldsymbol{H}' \in \mathbb{R}^{d\times d}$

$$\begin{aligned}
\|\boldsymbol{H} + \boldsymbol{H}'\| &= \sup_{\|\boldsymbol{x}\|\le 1} \|(\boldsymbol{H} + \boldsymbol{H}')\boldsymbol{x}\|_* \le \sup_{\|\boldsymbol{x}\|\le 1} (\|\boldsymbol{H}\boldsymbol{x}\|_* + \|\boldsymbol{H}'\boldsymbol{x}\|_*) \\
&\le \sup_{\|\boldsymbol{x}\|\le 1} \|\boldsymbol{H}\boldsymbol{x}\|_* + \sup_{\|\boldsymbol{x}'\|\le 1} \|\boldsymbol{H}'\boldsymbol{x}'\|_* = \|\boldsymbol{H}\| + \|\boldsymbol{H}'\|.
\end{aligned} \tag{53}$$

Now assume $\|\nabla^2 f(\boldsymbol{x})\| \le L$ for all $x \in \mathbb{R}^d$. Let $\boldsymbol{x}, \boldsymbol{y} \in \mathbb{R}^d$ and define $g(\tau) = \nabla f(\boldsymbol{x} + \tau(\boldsymbol{y} - \boldsymbol{x}))$ for $\tau \in [0, 1]$.

$$\begin{aligned}
\|\nabla f(\boldsymbol{y}) - \nabla f(\boldsymbol{x})\|_* &= \left\|\int_0^1 \frac{d}{d\tau} g(\tau) d\tau\right\|_* \\
&= \left\|\int_0^1 \nabla^2 f(\boldsymbol{x} + \tau(\boldsymbol{y} - \boldsymbol{x}))(\boldsymbol{y} - \boldsymbol{x}) d\tau\right\|_* \\
&\le \int_0^1 \left\|\nabla^2 f(\boldsymbol{x} + \tau(\boldsymbol{y} - \boldsymbol{x}))(\boldsymbol{y} - \boldsymbol{x})\right\|_* d\tau \\
&\stackrel{(52)}{\le} \int_0^1 \|\nabla^2 f(\boldsymbol{x} + \tau(\boldsymbol{y} - \boldsymbol{x}))\|\|\boldsymbol{y} - \boldsymbol{x}\| d\tau \\
&\le L\|\boldsymbol{y} - \boldsymbol{x}\| \int_0^1 1 d\tau = L\|\boldsymbol{y} - \boldsymbol{x}\|.
\end{aligned} \tag{54}$$

Conversely, assume $L$-smoothness and fix $\boldsymbol{x} \in \mathbb{R}^d$. For any $\|\boldsymbol{s}\| \le 1$ and $\varepsilon > 0$,

$$\left\|\left(\int_0^\varepsilon \nabla^2 f(\boldsymbol{x} + \tau\boldsymbol{s}) d\tau\right)\boldsymbol{s}\right\|_* = \|\nabla f(\boldsymbol{x} + \varepsilon\boldsymbol{s}) - \nabla f(\boldsymbol{x})\|_* \le \varepsilon L\|\boldsymbol{s}\| \le \varepsilon L. \tag{55}$$

Dividing by $\varepsilon$ and letting $\varepsilon \to 0$, we get

$$\begin{aligned}
\|\nabla^2 f(\boldsymbol{x})\boldsymbol{s}\|_* &= \left\|\lim_{\varepsilon\to 0}\left(\frac{1}{\varepsilon}\int_0^\varepsilon \nabla^2 f(\boldsymbol{x} + \tau\boldsymbol{s}) d\tau\right)\boldsymbol{s}\right\|_* \\
&= \lim_{\varepsilon\to 0}\frac{1}{\varepsilon}\left\|\left(\int_0^\varepsilon \nabla^2 f(\boldsymbol{x} + \tau\boldsymbol{s}) d\tau\right)\boldsymbol{s}\right\|_* \le L.
\end{aligned} \tag{56}$$

This implies $\|\nabla^2 f(\boldsymbol{x})\| = \sup_{\|\boldsymbol{s}\|\le 1} \|\nabla^2 f(\boldsymbol{x})\boldsymbol{s}\|_* \le L$. $\qquad\square$

### E.4 PROOF OF PROPOSITION 3

*Proof.* First note that

$$\|\boldsymbol{H}\|_{\infty,1} := \sup_{\|\boldsymbol{x}\|_\infty\le 1} \|\boldsymbol{H}\boldsymbol{x}\|_1 = \sup_{\|\boldsymbol{x}\|_\infty\le 1} \sum_i \left|\sum_j H_{ij}x_j\right| \le \sum_{i,j} |H_{ij}|. \tag{57}$$

Recall that $\sum_i |H_{ii}| = \sum_i H_{ii} = \sum_i \lambda_i$ for positive definite matrices. Then

$$\|\boldsymbol{H}\|_{\infty,1} \le \sum_{i,j} |H_{ij}| = \frac{\sum_{i,j} |H_{ij}|}{\sum_i |H_{ii}|} \sum_i \lambda_i = \rho_{\mathrm{diag}}(\boldsymbol{H})^{-1} \sum_i \lambda_i. \tag{58}$$

$\square$

### E.5  PROOF OF EQ. (9)

*Proof.* The fact that $\|z\|_\infty \leq \|z\|_2 \leq \|z\|_1$ implies

$$L_2 = \sup_{x \neq y} \frac{\|\nabla f(y) - \nabla f(x)\|_2}{\|y - x\|_2} \leq \sup_{x \neq y} \frac{\|\nabla f(y) - \nabla f(x)\|_1}{\|y - x\|_\infty} = L_\infty. \tag{59}$$

Conversely, using $\frac{1}{\sqrt{d}}\|z\|_1 \leq \|z\|_2 \leq \sqrt{d}\|z\|_\infty$

$$L_\infty = \sup_{x \neq y} \frac{\|\nabla f(y) - \nabla f(x)\|_1}{\|y - x\|_\infty} \leq \sup_{x \neq y} \frac{\sqrt{d}\|\nabla f(y) - \nabla f(x)\|_2}{\frac{1}{\sqrt{d}}\|y - x\|_2} = dL_2. \tag{60}$$

$\square$

### E.6  PROOF OF LEMMA 3

*Proof.* Statement (a) follows immediately from the definition of the dual norm.

Regarding (b), by definition of $z^{\|\cdot\|}$, we know that $\langle z, cz^{\|\cdot\|}\rangle - \frac{1}{2}\|cz^{\|\cdot\|}\|^2$ is maximized by $c = 1$. Hence the derivative w.r.t. $c$,

$$\frac{d}{dc}\left[\langle z, cz^{\|\cdot\|}\rangle - \frac{1}{2}\|cz^{\|\cdot\|}\|^2\right] = \langle z, z^{\|\cdot\|}\rangle - c\|z^{\|\cdot\|}\|^2, \tag{61}$$

must evaluate to 0 at $c = 1$, which proves (b).

For $(c)$, we use the equivalent definition

$$\|z\|_* = \max_{x \neq 0} \frac{\langle x, z\rangle}{\|x\|}. \tag{62}$$

Assume w.l.o.g. that $z^{\|\cdot\|} \neq 0$. Then

$$\|z\|_* = \max_{x \neq 0} \frac{\langle x, z\rangle}{\|x\|} \geq \frac{\langle z^{\|\cdot\|}, z\rangle}{\|z^{\|\cdot\|}\|} \overset{(b)}{=} \|z^{\|\cdot\|}\|. \tag{63}$$

Conversely, Let $x' \in \arg\max_{\|x\|=1}\langle x, z\rangle$, such that $\langle z, x'\rangle = \|z\|_*$. Then

$$\frac{1}{2}\|z\|_*^2 = \langle z, \|z\|_* x'\rangle - \frac{1}{2}\|\|z\|_* x'\|^2 \leq \langle z, z^{\|\cdot\|}\rangle - \frac{1}{2}\|z^{\|\cdot\|}\|^2 \overset{(b)}{=} \frac{1}{2}\|z^{\|\cdot\|}\|^2, \tag{64}$$

where the inequality is by definition of $z^{\|\cdot\|}$. $\square$

### E.7  PROOF OF PROPOSITION 4

*Proof.* The dual norm of $\|\cdot\|_L$ is $\|\cdot\|_{L^{-1}}$, such that the assumption of 1-smoothness w.r.t. $\|\cdot\|_L$ amounts to

$$\|\nabla f(y) - \nabla f(x)\|_{L^{-1}} \leq \|y - x\|_L \quad \forall x, y \in \mathbb{R}^d. \tag{65}$$

First, by definition of the maximum norm, we get

$$\|z\|_L = \sqrt{\sum_i l_i z_i^2} \leq \sqrt{\sum_i l_i \|z\|_\infty^2} = \sqrt{\sum_i l_i}\|z\|_\infty. \tag{66}$$

Secondly, using Cauchy-Schwarz,

$$\|z\|_1 = \sum_i |z_i| = \sum_i \frac{|z_i|}{\sqrt{l_i}}\sqrt{l_i} \leq \sqrt{\sum_i \frac{z_i^2}{l_i}}\sqrt{\sum_i l_i} = \sqrt{\sum_i l_i}\|z\|_{L^{-1}}. \tag{67}$$

Combining these two inequalities with the assumption yields the assertion:

$$\|\nabla f(y) - \nabla f(x)\|_1 \overset{(67)}{\leq} \sqrt{\sum_i l_i}\|\nabla f(y) - \nabla f(x)\|_{L^{-1}} \overset{(65)}{\leq} \sqrt{\sum_i l_i}\|y - x\|_L$$

$$\overset{(66)}{\leq} \sqrt{\sum_i l_i}\sqrt{\sum_i l_i}\|y - x\|_\infty = \left(\sum_i l_i\right)\|y - x\|_\infty. \tag{68}$$

$\square$

## E.8  PROOF OF EQ. (15)

*Proof.* The equivalence arises from substituting $\boldsymbol{y} = -\frac{1}{\alpha_t}(\boldsymbol{x} - \boldsymbol{x}_t)$ or $\boldsymbol{x} = \boldsymbol{x}_t - \alpha_t \boldsymbol{y}$. With a slight abuse of notation:

$$\boldsymbol{x}_{t+1} \in \underset{\boldsymbol{x}}{\arg\min} \left( \langle \nabla f_t, \boldsymbol{x} - \boldsymbol{x}_t \rangle + \frac{1}{2\alpha_t} \|\boldsymbol{x} - \boldsymbol{x}_t\|^2 \right)$$

$$\Longleftrightarrow \boldsymbol{x}_{t+1} \in \boldsymbol{x}_t - \alpha_t \underset{\boldsymbol{y}}{\arg\min} \left( \langle \nabla f_t, -\alpha_t \boldsymbol{y} \rangle + \frac{1}{2\alpha_t} \| - \alpha_t \boldsymbol{y}\|^2 \right)$$

$$\Longleftrightarrow \boldsymbol{x}_{t+1} \in \boldsymbol{x}_t - \alpha_t \underset{\boldsymbol{y}}{\arg\min} \left( \alpha_t \left[ -\langle \nabla f_t, \boldsymbol{y} \rangle + \frac{1}{2} \|\boldsymbol{y}\|^2 \right] \right)$$

$$\Longleftrightarrow \boldsymbol{x}_{t+1} \in \boldsymbol{x}_t - \alpha_t \underset{\boldsymbol{y}}{\arg\max} \left( \langle \nabla f_t, \boldsymbol{y} \rangle - \frac{1}{2} \|\boldsymbol{y}\|^2 \right). \tag{69}$$

$\square$

## E.9  PROOF OF PROPOSITION 5

*Proof.* First inequality: First, let $\hat{l}_1, \ldots, \hat{l}_d \geq 0$ be the minimizer in the definition of $L_{\text{sep}}$. For any $\boldsymbol{z}$ with $\|\boldsymbol{z}\|_\infty \leq 1$, we have

$$\boldsymbol{z}^T \boldsymbol{H} \boldsymbol{z} \leq \boldsymbol{z}^T \text{diag}(\hat{l}_1, \ldots, \hat{l}_d) \boldsymbol{z} = \sum_i \hat{l}_i z_i^2 \leq \sum_i \hat{l}_i. \tag{70}$$

Next, we rewrite the definition of $L_\infty$ as

$$L_\infty = \max_{\|\boldsymbol{x}\|_\infty, \|\boldsymbol{y}\|_\infty \leq 1} \boldsymbol{x}^T \boldsymbol{H} \boldsymbol{y}. \tag{71}$$

and let $(\hat{\boldsymbol{x}}, \hat{\boldsymbol{y}})$ be the maximizer. Then due to $\boldsymbol{H}$ being psd, we have

$$0 \leq (\hat{\boldsymbol{x}} - \hat{\boldsymbol{y}})^T \boldsymbol{H} (\hat{\boldsymbol{x}} - \hat{\boldsymbol{y}}) = \hat{\boldsymbol{x}}^T \boldsymbol{H} \hat{\boldsymbol{x}} - 2\hat{\boldsymbol{x}}^T \boldsymbol{H} \hat{\boldsymbol{y}} + \hat{\boldsymbol{y}}^T \boldsymbol{H} \hat{\boldsymbol{y}} \leq 2 \sum_i \hat{l}_i - 2\hat{\boldsymbol{x}}^T \boldsymbol{H} \hat{\boldsymbol{y}}, \tag{72}$$

where the last inequality is due to Eq. (70) applied to $\hat{\boldsymbol{x}}$ and $\hat{\boldsymbol{y}}$. This proves the assertion, since $\sum_i \hat{l}_i = L_{\text{sep}}$ and $\hat{\boldsymbol{x}}^T \boldsymbol{H} \hat{\boldsymbol{y}} = L_\infty$ by definition.

Second inequality: We set $l_i = \sum_j |H_{ij}|$ and denote $\boldsymbol{L} = \text{diag}(l_1, \ldots, l_d)$. Then

$$[\boldsymbol{L} - \boldsymbol{H}]_{ii} \geq \sum_{j \neq i} |H_{ij}| \tag{73}$$

$$\sum_{j \neq i} |[\boldsymbol{L} - \boldsymbol{H}]_{ij}| = \sum_{j \neq i} |H_{ij}| \tag{74}$$

making $\boldsymbol{L} - \boldsymbol{H}$ diagonally dominant with non-negative diagonal elements, hence positive semi-definite. Therefore, $\boldsymbol{L}$ is in the solution space of the definition of $L_{\text{sep}}$. Now,

$$\sum_i l_i = \sum_{i,j} |H_{ij}| = \left( \frac{\sum_i |H_{ii}|}{\sum_{i,j} |H_{ij}|} \right)^{-1} \sum_i \lambda_i, \tag{75}$$

where we used $\sum_i |H_{ii}| = \sum_i H_{ii} = \sum_i \lambda_i$. $\square$

## E.10  PROOF OF EQ. (34)

*Proof.* In two dimensions, the constraint $\text{diag}(l_1, l_2) - \boldsymbol{H} \succeq 0$ can be written as

$$l_1 - a \geq 0 \text{ and } (l_1 - a)(l_2 - d) - b^2 \geq 0 \tag{76}$$

by the leading principle minors criterion for positive definiteness. All "boundary" solutions can be written as

$$l_1 = a + s, \quad l_2 = d + \frac{b^2}{s}, \quad s \geq 0. \tag{77}$$

Out of those, the one that minimizes $l_1 + l_2$ is easily found to be given by $s = |b|$, resulting in

$$l_1 = a + |b|, \quad l_2 = d + |b|. \tag{78}$$

$\square$

### E.11 Proofs for Appendix C

All proofs are closely following the ones given for the Euclidean norm in Zhang et al. (2019).

To prove Lemma 4, we first need the following Lemma, which allows us to control the growth of the gradient norm in the vicinity of a point $\boldsymbol{x} \in \mathbb{R}^d$.

**Lemma 5.** *Assume Eq. (41) holds and let $\boldsymbol{x}, \boldsymbol{y}$ with $\|\boldsymbol{y} - \boldsymbol{x}\| \leq \frac{1}{L^{(1)}}$. Then*

$$\|\nabla f(\boldsymbol{y})\|_* \leq 4 \left( \frac{L^{(0)}}{L^{(1)}} + \|\nabla f(\boldsymbol{x})\|_* \right). \tag{79}$$

*Proof.* Define $\boldsymbol{x}(\tau) = \boldsymbol{x} + \tau(\boldsymbol{y} - \boldsymbol{x})$ as well as $\boldsymbol{g}(\tau) = \nabla f(\boldsymbol{x}(\tau))$ with $\boldsymbol{g}'(\tau) = \nabla^2 f(\boldsymbol{x}(\tau))(\boldsymbol{y} - \boldsymbol{x})$. Then

$$
\begin{aligned}
\|\nabla f(\boldsymbol{x}(t))\|_* &= \left\| \nabla f(\boldsymbol{x}) + \int_0^t \boldsymbol{g}'(\tau) d\tau \right\|_* \\
&\leq \|\nabla f(\boldsymbol{x})\|_* + \int_0^t \|\boldsymbol{g}'(\tau)\|_* d\tau \\
&= \|\nabla f(\boldsymbol{x})\|_* + \int_0^t \|\nabla^2 f(\boldsymbol{x}(\tau))(\boldsymbol{y} - \boldsymbol{x})\|_* d\tau \\
&\overset{(52)}{\leq} \|\nabla f(\boldsymbol{x})\|_* + \underbrace{\|(\boldsymbol{y} - \boldsymbol{x})\|}_{\leq 1/L^{(1)}} \int_0^t \underbrace{\|\nabla^2 f(\boldsymbol{x}(\tau))\|}_{\leq L^{(0)} + L^{(1)} \|\nabla f(\boldsymbol{x}(\tau))\|_* \ (41)} d\tau \\
&\leq \|\nabla f(\boldsymbol{x})\|_* + \frac{1}{L^{(1)}} \int_0^t L^{(0)} + L^{(1)} \|\nabla f(\boldsymbol{x}(\tau))\|_* d\tau \\
&= \|\nabla f(\boldsymbol{x})\|_* + t\frac{L^{(0)}}{L^{(1)}} + \int_0^t \|\nabla f(\boldsymbol{x}(\tau))\|_* d\tau
\end{aligned}
\tag{80}
$$

Applying the integral form of Groenwall's inequality[7] yields

$$\|\nabla f(\boldsymbol{x}(t))\|_* \leq \|\nabla f(\boldsymbol{x})\|_* + t\frac{L^{(0)}}{L^{(1)}} + \int_0^t \left( \|\nabla f(\boldsymbol{x})\|_* + \tau\frac{L^{(0)}}{L^{(1)}} \right) \exp(t - \tau) d\tau. \tag{82}$$

We now specialize to $t = 1$ and upper-bound the integrand

$$
\begin{aligned}
\|\nabla f(\boldsymbol{y})\|_* &= \|\nabla f(\boldsymbol{x}(1))\|_* \\
&\leq \|\nabla f(\boldsymbol{x})\|_* + \frac{L^{(0)}}{L^{(1)}} + \int_0^1 \left( \|\nabla f(\boldsymbol{x})\|_* + \underbrace{\tau}_{\leq 1}\frac{L^{(0)}}{L^{(1)}} \right) \underbrace{\exp(1 - \tau)}_{\leq \exp(1) < 3} d\tau \\
&\leq \|\nabla f(\boldsymbol{x})\|_* + \frac{L^{(0)}}{L^{(1)}} + 3 \left( \|\nabla f(\boldsymbol{x})\|_* + \frac{L^{(0)}}{L^{(1)}} \right) \int_0^1 d\tau \\
&= 4 \left( \frac{L^{(0)}}{L^{(1)}} + \|\nabla f(\boldsymbol{x})\|_* \right).
\end{aligned}
\tag{83}
$$

$\square$

We can now approach the proof of Lemma 4.

---

[7] Groenwall's inequality says that if $u(t) \leq \alpha(t) + \int_{t_0}^t \beta(\tau)u(\tau)d\tau$ for continuous $u$ and $\beta$, then

$$u(t) \leq \alpha(t) + \int_{t_0}^t \alpha(\tau)\beta(\tau) \exp\left( \int_\tau^t \beta(r)dr \right) d\tau. \tag{81}$$

We apply it here with $u(t) = \|\nabla f(\boldsymbol{x}(t))\|_*$ and $\alpha(t) = \|\nabla f(\boldsymbol{x})\|_* + tL^{(0)}/L^{(1)}$ and $\beta(\tau) \equiv 1$.

*Proof of Lemma 4.* According to Taylor's theorem we have

$$f(\boldsymbol{y}) = f(\boldsymbol{x}) + \langle \nabla f(\boldsymbol{x}), \boldsymbol{y} - \boldsymbol{x} \rangle + \frac{1}{2} \langle \boldsymbol{y} - \boldsymbol{x}, \nabla^2 f(\boldsymbol{\xi})(\boldsymbol{y} - \boldsymbol{x}) \rangle \tag{84}$$

with some $\boldsymbol{\xi} \in \{\boldsymbol{x} + \tau(\boldsymbol{y} - \boldsymbol{x}) \mid \tau \in [0,1]\}$. We can bound the quadratic term as

$$\begin{aligned}
\langle \boldsymbol{y} - \boldsymbol{x}, \nabla^2 f(\boldsymbol{\xi})(\boldsymbol{y} - \boldsymbol{x}) \rangle &\overset{(24)}{\leq} \|\boldsymbol{y} - \boldsymbol{x}\| \|\nabla^2 f(\boldsymbol{\xi})(\boldsymbol{y} - \boldsymbol{x})\|_* \\
&\overset{(52)}{\leq} \|\boldsymbol{y} - \boldsymbol{x}\|^2 \|\nabla^2 f(\boldsymbol{\xi})\| \\
&\overset{(41)}{\leq} (L^{(0)} + L^{(1)} \|\nabla f(\boldsymbol{\xi})\|_*) \|\boldsymbol{y} - \boldsymbol{x}\|^2.
\end{aligned} \tag{85}$$

The first inequality is by definition of the dual norm (see Lemma 3). The second inequality is by construction of the induced matrix norm. The final inequality uses the relaxed smoothness assumption (Eq. 41).

Next, since $\|\boldsymbol{y} - \boldsymbol{x}\| \leq 1/L^{(1)}$ by assumption of Lemma 4, we know $\|\boldsymbol{\xi} - \boldsymbol{x}\| \leq \frac{1}{L^{(1)}}$. Lemma 5 thus gives us

$$\|\nabla f(\boldsymbol{\xi})\|_* \leq 4 \left( \frac{L^{(0)}}{L^{(1)}} + \|\nabla f(\boldsymbol{x})\|_* \right). \tag{86}$$

Plugging this back into Eq. (85) yields

$$\langle \boldsymbol{y} - \boldsymbol{x}, \nabla^2 f(\boldsymbol{\xi})(\boldsymbol{y} - \boldsymbol{x}) \rangle \leq (5L^{(0)} + 4L^{(1)} \|\nabla f(\boldsymbol{x})\|_*) \|\boldsymbol{y} - \boldsymbol{x}\|^2. \tag{87}$$

Using that in Eq. (84) proves the assertion. $\square$

Finally, we prove Theorem 3

*Proof of Theorem 3.* Using Lemma 4 with $\boldsymbol{x} = \boldsymbol{x}_t$ and $\boldsymbol{y} = \boldsymbol{x}_{t+1} = \boldsymbol{x}_t - \eta_t \nabla f_t^{\|\cdot\|}$ yields

$$f_{t+1} \leq f_t - \eta_t \langle \nabla f_t, \nabla f_t^{\|\cdot\|} \rangle + \frac{\eta_t^2}{2} (5L^{(0)} + 4L^{(1)} \|\nabla f_t\|_*) \|\nabla f_t^{\|\cdot\|}\|^2. \tag{88}$$

Recall from Lemma 3 that $\langle z, z^{\|\cdot\|} \rangle = \|z^{\|\cdot\|}\|^2 = \|z\|_*^2$ and, hence,

$$\begin{aligned}
f_{t+1} &\leq f_t - \eta_t \langle \nabla f_t, \nabla f_t^{\|\cdot\|} \rangle + \frac{\eta_t^2}{2} (5L^{(0)} + 4L^{(1)} \|\nabla f_t\|_*) \|\nabla f_t^{\|\cdot\|}\|^2 \\
&= f_t - \left( \eta_t - \frac{\eta_t^2}{2} (5L^{(0)} + 4L^{(1)} \|\nabla f_t\|_*) \right) \|\nabla f_t\|_*^2 \\
&= f_t - \frac{\|\nabla f_t\|_*^2}{2(5L^{(0)} + 4L^{(1)} \|\nabla f_t\|_*)}.
\end{aligned} \tag{89}$$

If $\varepsilon \leq \|\nabla f_t\|_* \leq L^{(0)}/L^{(1)}$, we get

$$f_{t+1} \leq f_t - \frac{\varepsilon^2}{18L^{(0)}} \tag{90}$$

If $\|\nabla f_t\|_* \geq L^{(0)}/L^{(1)}$, we get

$$\begin{aligned}
f_{t+1} &\leq f_t - \frac{\|\nabla f_t\|_*^2}{2(5L^{(0)} + 4L^{(1)} \|\nabla f_t\|_*)} \\
&= f_t - \frac{\|\nabla f_t\|_*}{10L^{(0)}/\|\nabla f_t\|_* + 8L^{(1)}} \\
&\leq f_t - \frac{\|\nabla f_t\|_*}{18L^{(1)}} \\
&\leq f_t - \frac{L^{(0)}}{18(L^{(1)})^2}
\end{aligned} \tag{91}$$

Hence,

$$f_{t+1} \leq f_t - \min\left\{ \frac{L^{(0)}}{18(L^{(1)})^2}, \frac{\varepsilon^2}{18L^{(0)}} \right\}. \tag{92}$$

Now assume that we have $T$ iterations with $\|\nabla f_t\|_* \geq \varepsilon$. Then

$$f_0 - f_* \geq f_0 - f_T = \sum_{t=0}^{T-1}(f_t - f_{t+1}) \geq T\min\left\{ \frac{L^{(0)}}{18(L^{(1)})^2}, \frac{\varepsilon^2}{18L^{(0)}} \right\}. \tag{93}$$

Rearranging yields

$$T \leq 18\frac{f_0 - f_*}{\min\left\{ \frac{L^{(0)}}{(L^{(1)})^2}, \frac{\varepsilon^2}{L^{(0)}} \right\}} = 18(f_0 - f_*)\max\left( \frac{L^{(0)}}{\varepsilon^2}, \frac{(L^{(1)})^2}{L^{(0)}} \right). \tag{94}$$

$\square$

