# OpenReview forum: "The Geometry of Sign Gradient Descent"
_ICLR.cc/2020/Conference — Reject_

### Official Review · AnonReviewer1 · 2019-10-22
**Official Blind Review #1**

**Rating:** 3

**Review:**

This paper analyzes the performance of sign gradient descent as a function of the “geometry” of the objective. They contrast the performance of sign GD to gradient descent. Both algorithms are steepest descent methods with respect to different norms. Overall, I do not think the insights claimed in the paper are especially novel. It is already known that the choice of a norm can have a significant impact on the speed of steepest descent but practically, the performance depends on quantities that are unknown such as the norm of the gradient over the specific trajectory of the algorithm. I do not find the arguments in section 4 especially convincing as they seem rather high-level to me. Instead I would find it more valuable if one could make a statement about the performance of signGD as a function of known quantities. One could perhaps start by analyzing a specific class of functions (quadratics, Polyak-Lojasiewicz functions, …).

Prior work
It seems to me that most results discussed in the paper are already known in the optimization community. The book by Boyd & Vandenberghe (cited by the authors) has an entire section on the “Choice of norm for steepest descent” where they indeed explain that the choice of the norm used for steepest descent can potentially have a dramatic effect on the convergence rate. Can the authors explain how they see their contribution compared to prior work? I could concede that the result of Proposition 3 is somewhat novel, although I think the authors should discuss prior work on axis-alignment, which dates back to
Becker, Sue, and Yann Le Cun. "Improving the convergence of back-propagation learning with second order methods." Proceedings of the 1988 connectionist models summer school. 1988.

Section 4
The discussion in section 4 is high-level and I don’t see any particular insight one gains over what’s already known (see again book by Boyd & Vandenberghe). As explained by the authors, the comparison between signGD and GD will depend on the trajectory being followed.

Stochastic setting
Can the authors comment on generalizing their results to a stochastic setting? Since the gradients have a larger norm when using the infinity norm, I would expect that the variance is also larger.

Experimental results
The empirical results on the neural networks do not seem especially convincing, and the authors do seem to acknowledge this. What aspect of the analysis or the experimental setup do you expect to be responsible for this? Have you considered optimizing smaller models first? (something in between the quadratic problem in section 5.1 and the neural net in section 5.2. Some sort of ablation study where one strips away various components (batch-norm, residual connections, ….) of a neural network would also be valuable.

Generalization
One aspect that might be worth investigating is the generalization ability of steepest descent for different norms. There is already some prior work for adaptive methods that could perhaps be generalized, see e.g.
Wilson, Ashia C., et al. "The marginal value of adaptive gradient methods in machine learning." Advances in Neural Information Processing Systems. 2017.


**Experience Assessment:**

I have published in this field for several years.

**Review Assessment: Checking Correctness Of Derivations And Theory:**

I carefully checked the derivations and theory.

**Review Assessment: Checking Correctness Of Experiments:**

I assessed the sensibility of the experiments.

**Review Assessment: Thoroughness In Paper Reading:**

I read the paper thoroughly.

---

> ### Author Response · Authors · 2019-11-11
> **Response to R1**
>
> Thank you for your review. We gather that your main criticism is (1) that you "do not think the insights claimed in the paper are especially novel", and (2) that the results are "rather high-level" and "depend on quantities that are unknown such as the norm of the gradient". We address these two larger points first and then respond to specific questions/comments below.
>
>
> (1) Novelty and discussion of prior work
>
> After reading your review, we agree with you that we need to do a better job at contrasting with prior work to make our contributions stand out. We will update the paper shortly. We want to summarize for you what we see as the main contributions of our work. We believe that these are substantial and warrant publication. We hope that you will reconsider your judgement of their novelty in light of our response.
>
> The main goal of the paper is to understand the properties of the objective function which determine the performance of sign gradient descent relative to gradient descent. We do not achieve a full characterization (which we acknowledge, see also next comment) but we make the following contributions towards this goal which can not be found in any prior work:
>
> - We show that the smoothness constant for any norm arises as a bound on the Hessian in an induced matrix norm, relating it to properties of the Hessian. This is, of course, known for the Euclidean norm but we are not aware of any published works considering the general case.
>
> - For the smoothness constant w.r.t. the maximum norm, which is pertinent to the performance of signGD, we show an upper bound based on axis alignment and the eigenvalues. This sensitivity of signGD to the axis alignment has never been discussed before.
>
> - Based on that, we compare of GD and signGD. This comparison is qualitative since, as you point out, it depends on the unknown gradient norms over the trajectory of the algorithms. We also verify it experimentally for quadratic functions.
>
> - We explain how our insights relate to the "separable smoothness" assumption made in prior works on sign(S)GD. We clarify its meaning by showing that the sensitivity to axis alignment is hidden in the l_i values. We further show that this assumption can be replaced with the (weaker) assumption of max-norm smoothness in all existing results on sign(S)GD. I.e., we relax the assumptions of existing works while also clarifying the properties of the objective encoded in the assumption.
>
> With regards to the specific works you mentioned in your review:
> - Thanks for the pointer to Becker & Le Cun (1988), which supports our point that axis-alignment is a feature of neural network training objectives. We will add a brief discussion to the paper. Their interest in axis alignment stems from diagonal approximations in second order methods, which is an orthogonal direction.
> - Boyd & Vanderberghe indeed discuss how the choice of norm affects the convergence of steepest descent methods. However, the chapter “choice of norm for steepest descent” exclusively discusses quadratic norms resulting in preconditioned gradient descent methods.
>
>
> (2) Inconclusiveness of the results.
>
> We concede that the results in Section 4 are qualitative more than quantitative. As you rightly point out, this is because in addition to the smoothness constants, the performance depends on the gradient norms over the trajectory, which we can't get non-trivial lower bounds on. We nevertheless believe that understanding the smoothness constant gives important insights. We also verify this experimentally on quadratic functions.

---

> > ### Author Response · Authors · 2019-11-11
> > **Response to R1 ctd.**
> >
> > (3) Analyzing the stochastic setting
> >
> > We would like to note again that the main point of the paper is *not* any specific convergence statement, but rather to understand the properties that determine the L-inf smoothness constant, which will critically affect the performance of sign gradient methods in the deterministic as well the stochastic setting. As we point out in Section 6.2, the (weaker) max-norm smoothness can replace the separable smoothness assumption used in previous analyses of sign-based methods in the stochastic setting. I.e., we relax the assumption of existing works while also clarifying the properties of the objective encoded in the assumption.
> >
> > Porting a general convergence analysis of steepest descent to the stochastic setting would be an interesting endeavor, but poses many technical difficulties and is beyond the scope of this paper. Your comment regarding the gradient variance actually pertains to one of the main difficulties: In general it will not be possible to obtain an *unbiased* estimate of the steepest descent direction. If $g$ is an unbiased stochastic estimate of $\nabla f$, then $\Vert g\Vert_1 \text{sign}(g)$ will not be an unbiased estimate of $\Vert\nabla f\Vert_1\text{sign}(\nabla f)$. The error due to stochasticity thus has a bias and a variance component.
> >
> >
> > (4) Experiments.
> >
> > As we say in the paper, the empirical results on neural nets have to be taken with a grain of salt. Nevertheless, we think that they are convincing in that between the two models there is a considerable gap between in the observed improvement ratio, R(x_t), which is in line with the actual relative performance of GD/signGD on the two models.
> >
> >
> > (5) Generalization
> >
> > This is certainly an interesting direction but we think that it is best left for future work. Understanding sign(S)GD purely as an optimization method is an important step and adequate for scope of a single conference paper.

---

### Official Review · AnonReviewer3 · 2019-10-23
**Official Blind Review #3**

**Rating:** 3

**Review:**

This paper studies sign gradient descent in the framework of steepest descent with respect to a general norm. The authors show the connection to a smoothness with respect to a general norm, which is further related to a bound on the Hessian. Based on the interpretation of steepest descent, the authors compare the behavior of sign gradient descent with the gradient descent, and show that the relative behavior depends on both the percentage of diagonal ? and the spread of eigenvalues. Some experimental results are reported to verify the theory.

The paper is very well written and easy to follow. The theoretical analysis is sound and intuitive. However, the paper considers a variant of sign gradient descent where the l_1 norm of the gradients are introduced. The introduction of this l_1 norm is required for the interpretation of a steep descent but is not the one used often in practice.

Most analysis seem follows from standard arguments. The extension from gradient descent to the steep descent with a general norm is a bit standard. In this sense, the paper is a bit incremental.

----------------------
After rebuttal:

I have read the authors' response. I think the contribution may not be sufficient. I would like to keep my original score.

**Experience Assessment:**

I have read many papers in this area.

**Review Assessment: Checking Correctness Of Derivations And Theory:**

I did not assess the derivations or theory.

**Review Assessment: Checking Correctness Of Experiments:**

I did not assess the experiments.

**Review Assessment: Thoroughness In Paper Reading:**

I read the paper at least twice and used my best judgement in assessing the paper.

---

> ### Author Response · Authors · 2019-11-11
> **Response to R3**
>
> (1) "However, the paper considers a variant of sign gradient descent where the l_1 norm of the gradients are introduced."
>
> That is correct and the paper is completely open about that limitation; see footnote 2 in the paper. This allows us to leverage the gradient descent framework and also makes signGD converge with a constant step size to begin with. We believe that there's still a lot of insight to be gained from this analysis.
>
> This rescaling of the update direction can be rolled into the step size. In optimization it is fairly common to analyze a method with a special choice of step size (e.g., with exact line search) even if this deviates from practical choices. In practice, sign(S)GD is used with a decreasing step size schedule which can be seen as emulating the effect of the decreasing L1-norm of the gradient. Thus, although they are different algorithms, we do not expect the behaviour to be drastically different.
>
> In addition to all of our results,  we explore a possible direction to understand the normalization aspect of sign gradient descent via a relaxed smoothness condition in the appendix.
>
>
> (2) "The extension to [...] a general norm is a bit standard. In this sense, the paper is a bit incremental."
>
> We are astonished by this comment. Yes, this is absolutely standard, but this is *entirely* besides the point of the paper. The steepest descent framework is not the subject matter of the paper, it is merely a tool that we use to analyze sign gradient descent. The novelty lies in understanding the properties determining the performance of sign gradient descent, which we do by analyzing the non-Euclidean smoothness constant arising from the steepest descent framework.

---

### Official Review · AnonReviewer2 · 2019-10-23
**Official Blind Review #2**

**Rating:** 1

**Review:**

The paper tries to study the sign gradient descent by the squared maximum norm.  The authors clarify the meaning of a certain separable smoothness assumption using previous studies for signed gradient descent methods.

1. The authors change the problem. They study the sign gradient times its norm, not the classical sign gradient.
In fact, this change dramatically changes the flow in continuous time.

2. The results are too general, which does not focus on machine learning problems.

3. The paper is clearly not written well with many typos. The organization of the paper also needs to be improved a lot for publication.

For these reasons, I clearly reject this paper for publications.

**Experience Assessment:**

I have read many papers in this area.

**Review Assessment: Checking Correctness Of Derivations And Theory:**

I assessed the sensibility of the derivations and theory.

**Review Assessment: Checking Correctness Of Experiments:**

I assessed the sensibility of the experiments.

**Review Assessment: Thoroughness In Paper Reading:**

I read the paper at least twice and used my best judgement in assessing the paper.

---

> ### Author Response · Authors · 2019-11-11
> **Response to R2**
>
> (1) "They study the sign gradient times its norm, not the classical sign gradient. "
>
> That is correct and the paper is completely open about that limitation; see footnote 2 in the paper. This allows us to leverage the gradient descent framework and also makes signGD converge with a constant step size to begin with. We are thus baffled by your claim that we “change the problem” as there's still a lot of insight to be gained from this analysis.
>
> This rescaling of the update direction can be rolled into the step size. In optimization it is fairly common to analyze a method with a special choice of step size (e.g., with exact line search) even if this deviates from practical choices. In practice, sign(S)GD is used with a decreasing step size schedule which can be seen as emulating the effect of the decreasing L1-norm of the gradient. Thus, although they are different algorithms, we do not expect the behaviour to be drastically different.
>
> In addition to all of our results,  we explore a possible direction to understand the normalization aspect of sign gradient descent via a relaxed smoothness condition in the appendix.
>
>
> (2) "The results are too general and do not focus on ML."
>
> We analyze a method that is popular in ML. We view the fact that this analysis is kept as general as possible (and as specific as necessary) as a quality, not a drawback.

---

### Decision · Program_Chairs · 2019-12-19

**Decision:**

Reject

**Comment:**

The paper is rejected based on unanimous reviews.